# STATE-AWARE NEURAL STOCHASTIC DIFFERENTIAL EQUATIONS FOR MULTI-MODAL DYNAMICS

## ABSTRACT

Neural Stochastic Differential Equations (NSDEs) have recently attracted wide attention as a promising tool for modeling dynamical systems. However, we find that existing NSDE frameworks struggle to capture multimodal distributions, which are prevalent in real-world scenarios. To better understand this failure, we analyze the origins of multimodality in real-world data and show that it largely arises from shifts in the underlying data-generating process (DGP). We then further provide a theoretical explanation for why current NSDEs fail in these scenarios. To address this fundamental limitation, we propose the Multimodal Neural SDE (MM-NSDE) framework. MM-NSDE automatically perceives shifts in the underlying DGP and adaptively modifies the SDE dynamics, enabling more effective modeling of multimodal behaviors. Experiments on both synthetic and real-world datasets demonstrate that MM-NSDE achieves stable state-of-the-art performance. Remarkably, MM-NSDE is highly parameter-efficient, surpassing Mamba's performance while using only 1% of its parameter count. To facilitate further research, we release the code and implementation details at the following link: `https://anonymous.4open.science/r/MMNSDE-10EF`.

## 1 INTRODUCTION

Neural Stochastic Differential Equations (NSDEs) (Tzen and Raginsky, 2019) extend classical SDEs (Black and Scholes, 1973; Merton, 1973; Yang et al., 2020; Mariani et al., 2022) by parameterizing drift and diffusion with neural networks, thereby removing the need for predefined functional assumptions. Compared to deterministic sequence models (Rumelhart et al., 1986; Hochreiter and Schmidhuber, 1997) or state space models (Rangapuram et al., 2018; Forgione and Piga, 2023; Li et al., 2021), NSDEs naturally capture both continuous-time dynamics and stochasticity, making them a principled choice for modeling complex dynamical systems. However, we observe that NSDEs often fail to capture multimodal terminal distributions, despite the absence of such an assumption as a priori. Even in synthetic settings specifically designed to induce them, our experiments (Section 4.2) show that standard NSDEs rarely generate clearly multimodal behaviors.

To understand this failure, we first analyze the sources of multimodality in real-world data (Jalali et al., 2023). As illustrated in Figure 1, multimodal outcomes frequently arise from switching in the underlying data-generating process (DGP). For example, financial markets alternate between bullish and bearish phases, while physiological signals such as blood glucose exhibit recurrent transitions between active and resting states. We then provide a theoretical explanation for why NSDEs cannot adequately represent such distributions. The central issue is a Lipschitz conflict: training objectives in push-forward models (Arjovsky et al., 2017) require small Lipschitz constants to guarantee stability and convergence; for NSDEs, such constraints are also needed to ensure the existence and uniqueness of solutions. In contrast, capturing abrupt transitions in the data-generating process demands large Lipschitz constants to represent highly non-smooth dynamics. This trade-off substantially limits the expressive capacity of standard NSDEs in multimodal scenarios.

Motivated by these limitations, we propose MultiModal NSDEs (MM-NSDEs). MM-NSDE introduces a state-aware module that detects switching in the underlying data-generating process and updates the NSDE formulation accordingly. This self-adaptive design preserves the ability of NSDEs to model a single DGP, while also adapting to high-frequency or even continuous switching scenarios. In summary, our contributions are threefold:

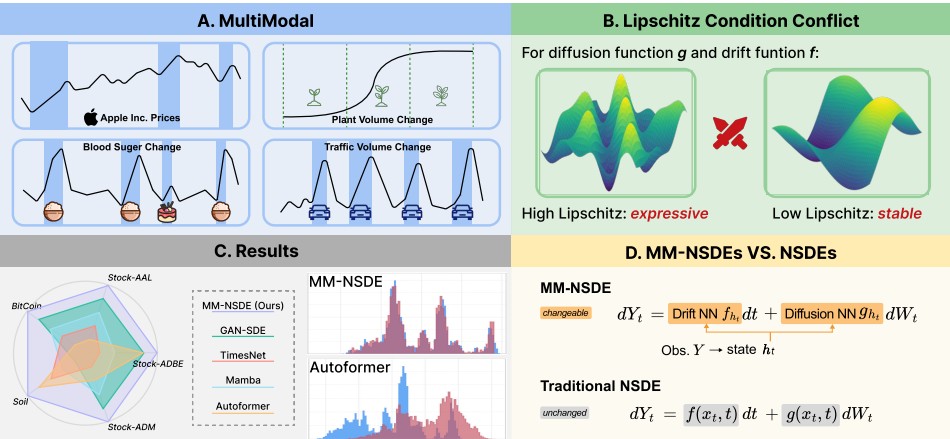

Figure 1: Motivation and overview of MM-NSDE. (A) Real-world time series (*e.g.*, bull vs. bear markets, plant growth stages, blood pressure, and traffic flow) naturally exhibit multimodal terminal distributions. In practice, sliding-window segmentation mixes regimes into the training set, leading to aggregated multimodality. (B) Standard NSDEs face a Lipschitz conflict: high constants ensure expressiveness but harm stability, while low constants stabilize at the cost of limited expressiveness. (C) Empirical results on Bitcoin dataset, show MM-NSDE consistently outperforms strong baselines. (D) Unlike traditional NSDEs, MM-NSDE employs adaptive drift and diffusion networks, avoiding the Lipschitz conflict and enabling robust multimodal modeling.

- We analyze the sources of multimodality in real-world data and explain why standard NSDEs fail to capture this ubiquitous phenomenon, pinpointing the Lipschitz conflict in both drift and diffusion terms as the key limiting factor.
- We propose **MultiModal NSDEs (MM-NSDEs)**, an elegant and effective extension that introduces a state-aware module to adaptively update the NSDE formulation in response to switching behaviors.
- We conduct extensive experiments on both synthetic and real-world datasets, demonstrating that MM-NSDE consistently outperforms existing approaches while maintaining parameter efficiency. In addition, we provide a detailed theoretical analysis to further support our method.

## 2 RELATED WORK

Our related work is organized into three categories of models: NSDEs, state space models and other deep sequential models.

**NSDEs.** NSDEs (Tzen and Raginsky, 2019; Liu et al., 2019) are able to explicitly model system noise in continuous time, which makes them widely applied in finance (Cuchiero et al., 2020), generative modeling (Song et al., 2020), and scientific computing (Rackauckas et al., 2020). Compared with other models, their advantage lies in naturally capturing stochastic dynamics. However, existing NSDEs face a fundamental trade-off: ensuring training stability, maintaining uniqueness of solutions, and achieving strong expressive power cannot be simultaneously guaranteed (Kidger et al., 2021; Li et al., 2020). This limitation prevents reliable modeling of multimodal trajectories.

**State Space Models.** To address DGP switching, another line of work introduces "state" as an explicit variable. From the classical Kalman Filter (Kalman, 1960) to modern deep variants such as RNNs (Gu et al., 2021; Smith et al., 2022) and Mamba (Gu and Dao, 2023), these models emphasize capturing long-range dependencies and state-aware modeling. Mamba further introduces structured state transitions to partially model regime changes explicitly. However, these approaches rely on discretization formulations and often assume Gaussian noise in structured equations, which limits their expressive power under highly non-stationary dynamics.

**Other Sequential models.** Recent architectures such as DLinear (Zeng et al., 2023), TimesNet (Wu et al., 2022), SegRNN (Lin et al., 2023), and Transformers (Liu et al., 2022) improve forecasting through structural innovations. Yet, they assume deterministic data generation and collapse to averages in stochastic regimes. While we acknowledge that such deterministic networks can serve as useful components in NSDE frameworks (e.g., as drift or diffusion modules), applying them directly in highly stochastic contexts leads to severe performance degradation.

## 3 METHODOLOGY

In this section, we first introduce NSDEs together with a formal definition of multimodal distribution, and then explain why existing approaches fall short in real-world scenarios. Building on this, Section 3.2 analyzes the root causes as DGP switching. And then Section 3.3 presents MM-NSDE with two key components—state-awareness and state-adaptive SDEs. Notably, additional explanation is given in Appendix 4, and complete proofs are deferred to Appendix 4.

### 3.1 PRELIMINARIES

**NSDEs.** We consider a $d$-dimensional stochastic process $\{Y_t\}_{t \geq 0}$. In traditional SDEs, the drift and diffusion functions, $f_t(Y_t)$ and $g_t(Y_t)$, are typically assumed to be analytically tractable. NSDEs instead leverage neural networks as flexible approximators:

$$dY_t = f_{\theta_1,t}(Y_t)\,dt + g_{\theta_2,t}(Y_t)\,dW_t, \tag{1}$$

where, for each $t \geq 0$, $f_{\theta_1,t} : \mathbb{R}^d \to \mathbb{R}^d$ and $g_{\theta_2,t} : \mathbb{R}^d \to \mathbb{R}^{d \times m}$ are neural networks with parameters $\theta_1, \theta_2$. The function $f_{\theta_1,t}$ specifies the drift term that describes the average dynamics of the system at time $t$, and $g_{\theta_2,t}$ specifies the diffusion term that accounts for stochastic fluctuations. Here, $W_t$ is an $m$-dimensional Wiener process that drives the stochasticity of the system. In the special case $m = d$, each state dimension is driven by an independent noise source, whereas when $m = 1$, a single noise source is shared across all dimensions.

**Multimodal Distribution.** At any fixed time $t$, the random variable $Y_t \in \mathbb{R}^d$ is characterized by a transition probability density $Y_t \sim p(\cdot \mid Y_0)$, $\quad p : \mathbb{R}^d \to \mathbb{R}_+$. A fundamental challenge for NSDEs arises when this transition distribution is *multimodal* rather than unimodal. Intuitively, multimodality means that the density is concentrated in multiple distinct regions (or "peaks"), which often correspond to shifts in the underlying data-generating process.

> **Definition 1** ($K$-modality in $\mathbb{R}^d$). *Let $p : \mathbb{R}^d \to \mathbb{R}_+$ be a continuous probability density function. If there exists a threshold $\lambda_\star \in (0, \sup p)$ such that the super-level set*
> $$L(\lambda_\star) := \{y \in \mathbb{R}^d \mid p(y) \geq \lambda_\star\}$$
> *has exactly $K$ connected components, then we say that $p$ is $K$-modal. If $K = 1$, the distribution is called unimodal; If $K > 1$, the distribution is called multimodal.*

### 3.2 FAILURE ANALYSIS OF NSDES.

There are two widely recognized reasons why NSDEs are often designed with relatively small Lipschitz constants. From classical SDE theory (Øksendal, 2003; Karatzas and Shreve, 1991), the drift and diffusion functions are required to be Lipschitz continuous in order to guarantee the existence and uniqueness of strong solutions. This condition naturally extends to neural parameterizations of SDEs. In particular, SDE-Net (Liu et al., 2019) established the following guarantee:

> **Theorem 2** (Existence and Uniqueness of Neural SDE Solutions (Liu et al., 2019)). *Suppose there exists a constant $C > 0$ such that for all $y, z \in \mathbb{R}^d$ and $t \geq 0$,*
> $$\|f_{\theta_1,t}(y) - f_{\theta_1,t}(z)\| + \|g_{\theta_2,t}(y) - g_{\theta_2,t}(z)\| \leq C\|y - z\|.$$
> *Then, for every $Y_0 \in \mathbb{R}^d$, there exists a unique continuous and adapted process $\{Y_t\}_{t \geq 0}$ satisfying*
> $$Y_t = Y_0 + \int_0^t f_{\theta_1,s}(Y_s)\,ds + \int_0^t g_{\theta_2,s}(Y_s)\,dW_s,$$
> *and moreover, $\mathbb{E}\big[\sup_{0 \leq s \leq T} \|Y_s\|^2\big] < +\infty$ for all $T \geq 0$.*

Beyond existence and uniqueness, small Lipschitz constants are also favored in practice for stabilizing training. Empirical evidence from generative modeling suggests that excessive Lipschitz constants often lead to exploding gradients or unstable dynamics. Consequently, many works impose Lipschitz control through techniques such as spectral normalization or gradient penalties. Although most of these observations were originally made in the context of GANs and VAEs, the same principle applies to NSDEs: bounding the Lipschitz constant of $f_{\theta_1}$ and $g_{\theta_2}$ helps maintain numerically stable trajectories and prevents mode collapse during training.

While small Lipschitz constants guarantee stability, Theorem 3 makes clear that they also suppress the amplification factor $\mathcal{A}_t$. Because the threshold for separated multimodality grows exponentially in $\delta/\sigma$, dynamics that are too contractive cannot reach it. A NSDE with small $L_f$ and $L_g$ may lacks the expressive capacity to generate separated multimodality, even under state-dependent diffusion.

> **Theorem 3** (Necessary amplification for separated multimodality). *If the terminal law $\nu = \mathcal{L}(Y_t)$ exhibits separated bimodality in some direction with mixture weight $\lambda$, separation $\delta > 0$, and scale $\sigma > 0$, then there exist constants $c_0, c_1 > 0$ (depending only on dimension and ellipticity) such that*
>
> $$\|g\|_\infty \exp\big(c_1(L_f + L_g^2)t\big) \ \geq \ c_0^{-1}\,\sigma \exp\Big(\tfrac{\delta^2}{8\sigma^2} - \tfrac{1}{2}(\Phi^{-1}(\lambda))^2\Big),$$
>
> *where $L_f$ and $L_g$ denote the Lipschitz constants of the drift and diffusion in $y$, respectively, and $\|g\|_\infty = \sup_{(y,s)} \|g_{\theta_2,s}(y)\|_{\mathrm{op}}$.*

The stability of standard NSDEs typically relies on strict Lipschitz constraints imposed on the drift and diffusion functions (see Theorem 2). On the one hand, such constraints guarantee existence and uniqueness of the solution and ensure stable training; on the other hand, they significantly limit the ability of the system to generate separated multimodal distributions within a finite horizon (see Theorem 3). This reveals a **fundamental conflict**:

- **Prioritizing stability**: small Lipschitz constants ensure well-posedness, but a single NSDE lacks sufficient expressiveness to capture multimodal dynamics.
- **Prioritizing expressiveness**: relaxing the Lipschitz constraint increases modeling capacity but often leads to exploding gradients or unstable trajectories during training.

### 3.3 MM-NSDEs

**Problem Definition.** To balance the conflict, we introduce a Stare Awareness and adaptation mechanism, where our framework first identifies the current DGP and then applies an NSDE specialized for it. In this way, each NSDE only needs to model a single mode of the dynamics, thereby reducing the burden on expressiveness while maintaining stability.

This reformulation is also consistent with real-world scenarios: in practice, DGPs often change over time rather than remaining fixed, naturally giving rise to multimodal dynamics. More generally, we consider a multivariate time series $\{(\tau_n, y_n)\}_{n=0}^T$, $y_n \in \mathbb{R}^d$, $0 = \tau_0 < \tau_1 < \cdots < \tau_T$, possibly sampled at irregular timestamps and generated by an unknown DGP that may change over time. The task is to (i) recognize the current DGP and (ii) switch to the corresponding DGP to generate future trajectories at timestamps $\{\tau_{n+1}, \ldots, \tau_{n+H}\}$ with predictions $\{\hat{y}_{\tau_{n+1}}, \ldots, \hat{y}_{\tau_{n+H}}\}$. The predictive distribution is required to approximate the true conditional data distribution: $\hat{p}(Y_{[\tau_{n+1}:\tau_{n+H}]} \mid y_{[\tau_{n-k}:\tau_n]}) \approx p_{\mathrm{data}}(Y_{[\tau_{n+1}:\tau_{n+H}]} \mid y_{[\tau_{n-k}:\tau_n]})$, where $\hat{p}$ denotes the model-induced predictive distribution and $p_{\mathrm{data}}$ the true data distribution. In the autoregressive forecasting setting, this recognition and adaptation step must be performed at every prediction step: before generating $\hat{y}_{\tau_{n+m}}$, the system first infers the active DGP at time $\tau_{n+m-1}$ and then conditions the next-step dynamics accordingly.

**Stare Awareness Module.** To realize the proposed recognition mechanism, we introduce a Stare Awareness Module that infers the currently active DGP and embeds it into a continuous latent representation $h_t \in \mathbb{R}^D$. This latent process conditions the subsequent NSDE dynamics and evolves according to

$$dh_t = \underbrace{A_t\big(y_{[\tau_{n-k}:\tau_n]}\big) \odot h_t}_{\text{DGP transition}} dt + \underbrace{B_t\big(y_{[\tau_{n-k}:\tau_n]}\big)}_{\text{Input modulation}} dt,$$

where $y_{[\tau_{n-k}:\tau_n]} \in \mathbb{R}^{(k+1)\times d}$ denotes the most recent observation window of length $k+1$, and $h_t$ is the latent embedding of the active DGP, with the above dynamics applied for $t \in (\tau_n, \tau_{n+1}]$. The functions $A_t(\cdot)$ and $B_t(\cdot)$ are neural networks that take the observation window as input (the subscript $t$ indexes time), and $\odot$ denotes element-wise multiplication. The first term $A_t(\cdot) \odot h_t\,dt$ governs smooth transitions of the latent state, while the second term $B_t(\cdot)\,dt$ introduces input-driven corrections. In the case of a constant sampling interval $\Delta\tau = \tau_{n+1} - \tau_n$, the latent process admits the Euler discretization $h_{\tau_{n+1}} \approx h_{\tau_n} + \Delta\tau\big(A_{\Delta\tau}(y_{[\tau_{n-k}:\tau_n]}) \odot h_{\tau_n} + B_{\Delta\tau}(y_{[\tau_{n-k}:\tau_n]})\big)$.

**Stare Adaptive Module.** Once the active DGP is recognized, our framework needs to switch to the corresponding dynamics before performing prediction. Conditioned on the latent process $h_t$, we

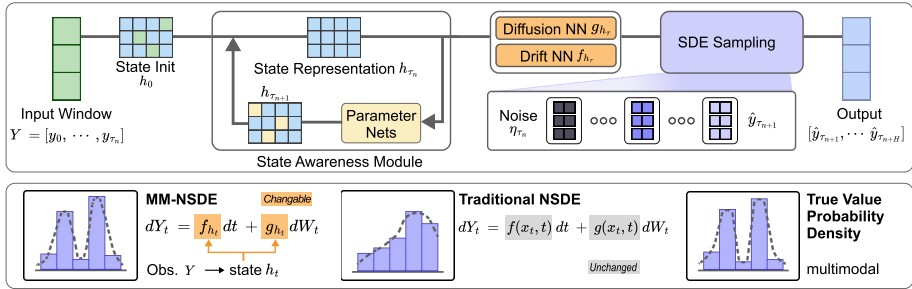

Figure 2: Overview of the MM-NSDE. A time-series window is mapped to a latent state by the State Awareness Module. The latent state then parameterizes the drift and diffusion of the State-Adaptive NSDE, which generates future trajectories.

transform the canonical drift and diffusion fields via modulators $H(\cdot)$ and $G(\cdot)$ generated from $h_t$. Specifically, let

$$\begin{cases} \tilde{f}_t(y) := H(h_t) \, f_{0,t}(y), \\ \tilde{g}_t(y) := G(h_t) \, g_{0,t}(y), \end{cases}$$

where, for each $t \geq 0$, $f_{0,t} : \mathbb{R}^d \to \mathbb{R}^d$ and $g_{0,t} : \mathbb{R}^d \to \mathbb{R}^{d \times m}$ are the base drift and diffusion functions. are base drift/diffusion functions. After modulation, the effective dynamics remain in the standard NSDE form: $dy_t = \tilde{f}_t(y_t) \, dt + \tilde{g}_t(y_t) \, dW_t$,. At discrete timestamps $\tau_n$ with step size $\Delta\tau_n = \tau_{n+1} - \tau_n$, the Euler–Maruyama update is

$$y_{\tau_{n+1}} \approx y_{\tau_n} + \tilde{f}_{\tau_n}(y_{\tau_n}) \, \Delta\tau_n + \tilde{g}_{\tau_n}(y_{\tau_n}) \, \eta_n \sqrt{\Delta\tau_n},$$

where $\eta_n \sim \mathcal{N}(0, I_m)$ denotes an $m$-dimensional standard Gaussian vector corresponding to the Brownian increments.

**Training Objective.** At each prediction step, our goal is to measure the discrepancy between the empirical distribution of predicted samples and that of the real data. A suitable loss must satisfy two criteria: (i) it should be *differentiable*, so that it can be integrated into gradient-based training, and (ii) it should be *sensitive to multimodality*, since real-world dynamics often involve multiple regimes or modes. To this end, we compare three loss functions: Maximum Mean Discrepancy (MMD), Wasserstein-2 distance, and the entropically regularized Sinkhorn divergence. Their sensitivity can be analyzed under a multimodal setting.

**Proposition 4** (Sensitivity to multimodality). *Under mild assumptions on the distributions (see Appendix X), we have the following asymptotic behavior as mode separation $a \to \infty$:*

$$W_2(p,q) = \Omega(a), \quad S_\varepsilon(p,q) = \Omega(a), \quad \mathrm{MMD}(p,q) = O(1).$$

Proposition 4 shows that MMD saturates and does not reflect increasing mode separation, while Wasserstein-2 and Sinkhorn remain sensitive to multimodality. We therefore adopt the entropically regularized Sinkhorn divergence, which combines this sensitivity with differentiability and efficiency. Concretely, consider a prediction horizon of length $H$ with future timestamps $\{\tau_{n+1}, \ldots, \tau_{n+H}\}$. Given a batch of real sequences $\{Y^{(i)}\}_{i=1}^N$, the model generates predictions $\{\hat{Y}^{(i)}\}_{i=1}^N$ in an autoregressive manner. At each step $m \in \{1, \ldots, H\}$, we define the empirical distributions $P_{\tau_{n+m}} = \frac{1}{N} \sum_{i=1}^N \delta_{y_{\tau_{n+m}}^{(i)}}$, $Q_{\tau_{n+m}} = \frac{1}{N} \sum_{i=1}^N \delta_{\hat{y}_{\tau_{n+m}}^{(i)}}$. The discrepancy between $P_{\tau_{n+m}}$ and $Q_{\tau_{n+m}}$ is measured by the Sinkhorn loss:

$$\mathcal{L}_{\tau_{n+m}} = \min_{\mathbf{T} \in \Pi(P_{\tau_{n+m}}, Q_{\tau_{n+m}})} \left[ \sum_{i,j} \mathbf{T}_{i,j} \, \|y_{\tau_{n+m}}^{(i)} - \hat{y}_{\tau_{n+m}}^{(j)}\|_2^2 + \lambda \cdot \mathrm{Entropy}(\mathbf{T}) \right],$$

where $\mathbf{T} \in \mathbb{R}_+^{N \times N}$ is constrained by $\Pi(P_{\tau_{n+m}}, Q_{\tau_{n+m}}) = \{\mathbf{T} : \mathbf{T}\mathbf{1} = \frac{1}{N}\mathbf{1}, \ \mathbf{T}^\top \mathbf{1} = \frac{1}{N}\mathbf{1}\}$, and the entropy regularizer is $\mathrm{Entropy}(\mathbf{T}) = -\sum_{i,j} \mathbf{T}_{i,j} \log \mathbf{T}_{i,j}$. This term avoids degenerate one-to-one matchings by encouraging smoother transport plans, stabilizing optimization and preserving information about all modes. The overall training objective is the average loss over the horizon.

## 4 EXPERIMENTS

In this section, we begin by outlining the experimental setup, including baselines, benchmarks, evaluation metrics (Section 4.1). We then present results on the simulated datasets (Section 4.2), followed by real-world datasets across finance, environment, and cryptocurrency (Section 4.3). Finally, we provide further analyses, including multimodality validation, sensitivity to input window length, scalability to high-dimensional data, and computational efficiency (Section 4.4).

### 4.1 EXPERIMENTAL SETUP

**Benchmarks.** We evaluate performance on simulated and real-world datasets. The simulated datasets are based on three representative SDEs: Geometric Brownian Motion (GBM), the Ornstein–Uhlenbeck process (OU), and the Cox–Ingersoll–Ross process (CIR). They fall into three categories: **1) Intra-family switching:** transitions occur within the same type of SDE. **2) Inter-family switching:** transitions occur across different SDE types. **3) Continuous switching:** parameters vary at each time step, capturing richer temporal dynamics. For real-world datasets, we use data from finance, cryptocurrency, and environmental domains. The financial data include stock prices of multiple companies, while the environmental dataset tracks daily municipal solid waste in five global cities. Further details are provided in Appendix 4.

**Baselines.** We compare MM-NSDEs with a broad set of representative baselines that cover both time-series forecasting and SDE-based generative modeling. For time-series forecasting, we consider a diverse set of architectures: DLinear (Zeng et al., 2023), a strong linear baseline; Seg-RNN (Lin et al., 2023), a recurrent model with segmentation capability; TimesNet (Wu et al., 2022), a convolutional architecture for capturing multi-periodic patterns; and the NS-Transformer (Zeng et al., 2023), a transformer-based model tailored for non-stationary forecasting. In addition, we include Mamba (Gu and Dao, 2023), a recent selective state-space model designed for efficient sequence modeling. For comparison with generative approaches, we adopt Latent-SDEs (Hauberg et al., 2023) and GAN-SDEs (Kidger et al., 2021), both of which leverage SDEs for sequence generation. Further implementation details and hyperparameter settings are given in Appendix 4.

**Task setting and Metrics.** For all sequence lengths, we use 100 points as historical input and 50 points as the prediction horizon. NSDEs are fundamentally generative models. Following prior work (Rhee and Glynn, 2015; Kidger et al., 2021), such models are capable of both conditional prediction and unconditional generation. Accordingly, we evaluate them using three metrics: the Mean Integrated Squared Error (MISE) and the difference in the tail cumulative distribution function (TD) for distributional fitting, and the Mean Squared Error (MSE) for prediction accuracy. Detailed formulas are provided in Appendix 4.

### 4.2 EVALUATION ON SIMULATED DATA

| Model | $GBM_1$ | | | | $OU_1$ | | | | $CIR_1$ | | | |
|---|---|---|---|---|---|---|---|---|---|---|---|---|
| | MISE↓ | TD↓ | MMD↓ | MSE↓ | MISE↓ | TD↓ | MMD↓ | MSE↓ | MISE↓ | TD↓ | MMD↓ | MSE↓ |
| DLinear | 0.21 | 0.02 | 0.08 | 1.01 | 1.21 | 0.09 | 0.52 | 1.21 | 1.38 | 0.09 | 0.46 | 1.16 |
| SegRNN | 0.53 | 0.07 | 0.14 | 2.01 | 0.36 | 0.08 | 0.09 | 1.72 | 0.59 | **0.01** | 0.13 | 1.49 |
| TimesNet | 0.61 | 0.08 | 0.15 | 2.10 | 0.37 | 0.01 | 0.08 | 1.74 | 0.60 | 0.03 | 0.13 | 1.51 |
| Autoformer | 0.12 | 0.08 | 0.17 | 2.06 | 0.36 | 0.02 | 0.09 | 1.81 | 0.56 | 0.03 | 0.09 | 1.80 |
| NS-Transformer | 0.19 | 0.04 | 0.05 | 1.13 | 1.42 | 0.00 | 0.07 | 1.75 | 0.87 | **0.01** | 0.08 | 1.14 |
| Mamba | 0.57 | 0.09 | 0.15 | 2.01 | 0.58 | 0.01 | 0.04 | 1.73 | 0.58 | **0.01** | 0.12 | 1.53 |
| Latent-SDEs | 0.97 | 0.10 | 0.19 | 2.35 | 0.68 | 0.09 | 0.48 | 1.60 | 0.89 | 0.08 | 0.41 | 1.36 |
| GAN-SDEs | 1.20 | 0.10 | 0.22 | 2.60 | 2.71 | 0.10 | 0.65 | 1.95 | 3.22 | 0.10 | 0.58 | 1.90 |
| MM-NSDEs | **0.04** | **0.01** | **0.12** | **0.75** | **0.08** | **0.00** | **0.03** | **0.02** | **0.08** | **0.01** | **0.03** | **0.21** |

Table 1: Comparison of different models on the simulated dataset with Intra-family switching. Our results are highlighted with darker shading, and the best performance is shown in bold. For prediction models, directly applying MSE supervision leads to mode collapse; therefore, we adopt the Sinkhorn loss as the training objective. For Latent-SDEs and GAN-SDEs, we rely on their publicly available implementations. Detailed experimental configurations are provided in the Appendix 4.

Our model addresses the stochastic simulation tasks, as shown in Table 1, achieving lower errors across all settings. In contrast, sequential prediction models are inadequate for stochastic sequence switching. With MSE supervision, they collapse to matching only second-order moments, leading to mode collapse. Even under distributional supervision, the learned trajectory law deviates from the ground truth. The error can be traced to three sources: discretization bias, distributional mismatch, and model capacity limitations. Any increase in these components amplifies the overall deviation,

causing models such as Mamba to exhibit large errors in our benchmarks. Further details of this analysis are provided in the Appendix. Latent-SDEs and GAN-SDEs fail for a different reason: Although these models have inherent stochasticity and appear capable of modeling such dynamics, Lipschitz conflicts in the drift and diffusion terms destabilize training and ultimately lead to collapse.

Figure 3 makes clear the capability of MM-NSDE. Although the continuous switching (Type-3) setting is difficult, as the underlying DGP changes at each time step, MM-NSDE successfully reproduces the evolving distributions. By contrast, most baseline models capture only the mean and overall range, and DLinear collapses entirely to the mean.

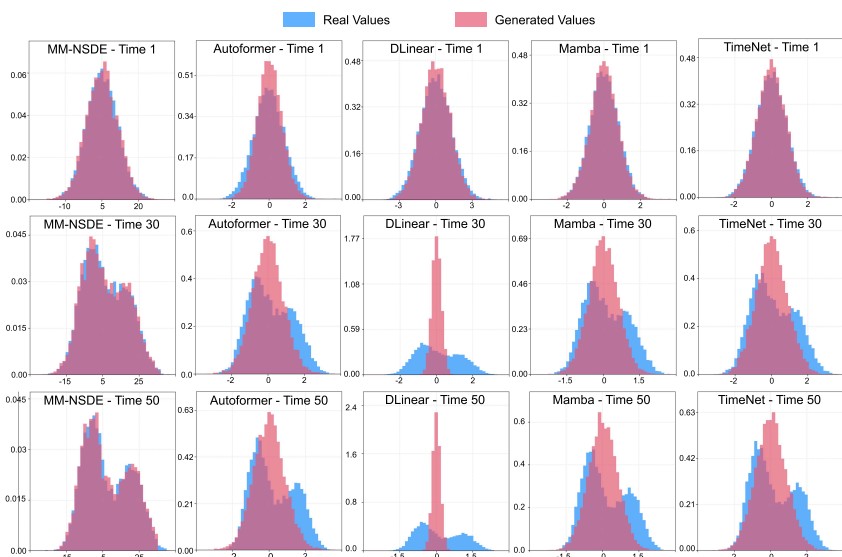

Figure 3: Comparison between the empirical distributions of real trajectories (blue) and generated trajectories (red) at different time steps for the continuous switching (Type-3) simulated data. Results are shown for MM-NSDE and four representative baseline models.

## 4.3 Evaluation on Real-world Data

| Model | Stock-AAL | | Stock-ADBE | | Stock-ADM | | Soil | | BitCoin | |
|---|---|---|---|---|---|---|---|---|---|---|
| | MISE↓ | TD↓ | MISE↓ | TD↓ | MISE↓ | TD↓ | MISE↓ | TD↓ | MISE↓ | TD↓ |
| Autoformer | 1.12 | 9.98 | 0.38 | 5.21 | 0.71 | 8.09 | 0.03 | 7.63 | 1.79 | 1.84 |
| DLinear | 1.59 | 10.00 | 1.33 | 8.80 | 1.69 | 9.03 | 0.64 | 0.71 | 1.58 | 8.78 |
| Mamba | 0.73 | 7.82 | 0.45 | 6.26 | 0.57 | 9.78 | 0.25 | 1.60 | 1.10 | 4.53 |
| NS-Transformer | 1.19 | 9.99 | 1.30 | 6.77 | 2.76 | 4.72 | 0.26 | 9.89 | 1.85 | 3.42 |
| SegRNN | 1.57 | 9.93 | 0.54 | 1.81 | 1.03 | 9.95 | 0.27 | 2.88 | 3.78 | 9.01 |
| TimesNet | 0.93 | 9.91 | 0.90 | 3.99 | 0.87 | 9.94 | 0.20 | 8.52 | 1.68 | 9.86 |
| Latent-SDE | 0.36 | 9.90 | 0.35 | 9.08 | 0.41 | 9.97 | 0.47 | 9.26 | 0.35 | 9.99 |
| GAN-SDE | 0.38 | 9.95 | 0.38 | 9.98 | 0.43 | 9.99 | 0.95 | 9.93 | 0.11 | 10.01 |
| MM-NSDE | **0.02** | **0.01** | **0.02** | **0.06** | **0.06** | **0.02** | **0.01** | **0.05** | **0.05** | **0.03** |

| Model | Stock-AAL | | Stock-ADBE | | Stock-ADM | | Soil | | BitCoin | |
|---|---|---|---|---|---|---|---|---|---|---|
| | MMD↓ | MSE↓ | MMD↓ | MSE↓ | MMD↓ | MSE↓ | MMD↓ | MSE↓ | MMD↓ | MSE↓ |
| Autoformer | 0.99 | 4.35 | 0.07 | 1.42 | 1.09 | 7.28 | 0.01 | 1.23 | 1.03 | 2.74 |
| DLinear | 0.72 | 3.59 | 1.09 | 4.41 | 1.13 | 8.69 | 0.30 | 1.25 | 0.99 | 5.32 |
| Mamba | 0.42 | 12.62 | 0.09 | 1.37 | 0.70 | 44.51 | 0.02 | 7.15 | 1.02 | 11.54 |
| NS-Transformer | 0.25 | 2.34 | 0.58 | 2.37 | 0.98 | 7.78 | 0.06 | 0.88 | 0.89 | 4.59 |
| SegRNN | 0.97 | 3.72 | 0.06 | 1.41 | 0.58 | 14.89 | 0.05 | 1.06 | 1.03 | 6.88 |
| TimesNet | 0.52 | 4.93 | 0.05 | 1.29 | 0.75 | 20.41 | 0.05 | 0.91 | 0.96 | 9.71 |
| Latent-SDE | 0.95 | 10.57 | 0.65 | 3.01 | 0.96 | 25.12 | 0.35 | 1.30 | 1.05 | 10.01 |
| GAN-SDE | 1.10 | 13.01 | 0.90 | 4.81 | 1.15 | 40.01 | 0.45 | 1.50 | 1.15 | 12.01 |
| MM-NSDE | **0.03** | **0.15** | **0.07** | **0.73** | **0.05** | **1.30** | **0.01** | **0.33** | **0.01** | **0.40** |

Table 2: Comparison of different models on various domains, including finance, environment, and bitcoin. All values are scaled by $10^{-2}$. Bold text indicates the best performance. ↓: lower is better.

From Table 2, we observe that sequential models perform poorly on volatile and non-stationary sequences such as stocks and Bitcoin. Their performance degrades severely in both error and distributional metrics; although Latent-SDE and GAN-SDE mitigate the issue, they cannot fully resolve

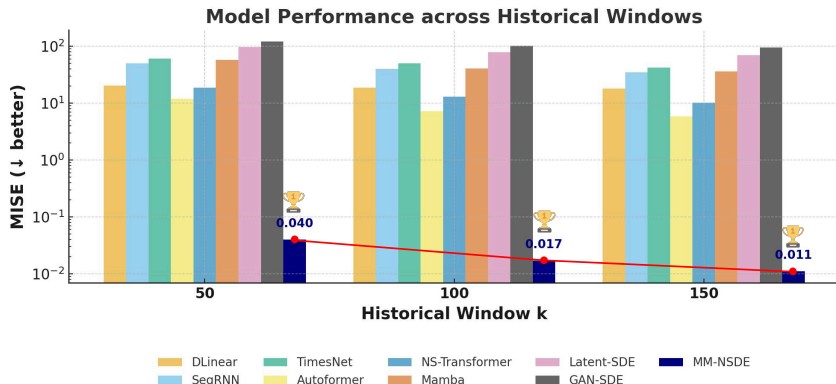

Figure 5: Model performance across varying historical window sizes $k \in \{50, 100, 150\}$, evaluated by MISE (lower is better). To examine the impact of different historical window sizes, all models are evaluated on the **GBM1 dataset** with sequence length 250, where a regime switch occurs at time step $t = 200$. All other experimental configurations remain identical to the previous setup.

it. On low-noise data such as Soil, traditional models achieve reasonable results, yet MM-NSDE still delivers the best performance across all metrics. Interestingly, even within the same financial market, different stocks demonstrate substantial variability in stochasticity, underscoring the need for models that can adapt to heterogeneous noise levels. We also observe that real-world data often display multi-modal behaviors that are more complex than simulation, where transformers tend to collapse into trivial solutions by directly copying the previous ground-truth value as the next-step prediction, highlighting their fundamental inability to capture stochasticity. MM-NSDE avoids such overfitting through its intrinsic stochasticity, while training each DGP with sufficient expressive capacity to capture multi-modal distributions.

### 4.4 FURTHER ANALYSIS

**Empirical Validation of Separated Multimodality.** We empirically validate Theorem 3, which states that small Lipschitz constants $(L_f, L_g)$ suppress the amplification factor $\mathcal{A}_t$, preventing the emergence of separated multimodality. To test this, we imposed *hard Lipschitz constraints* by applying spectral normalization to each linear layer (enforcing 1-Lipschitz) and scaling the outputs to the target constants $L_f$ and $L_g$. NSDEs were trained on a bimodal Gaussian mixture target using an MMD loss, and the terminal distributions were evaluated with a Gaussian mixture model. The separation index $\delta/\sigma > 2$ was taken as evidence of separated multimodality. As shown in Figure 4, separated multimodality appears only when $(L_f, L_g)$ are sufficiently large, while smaller values fail to reach the

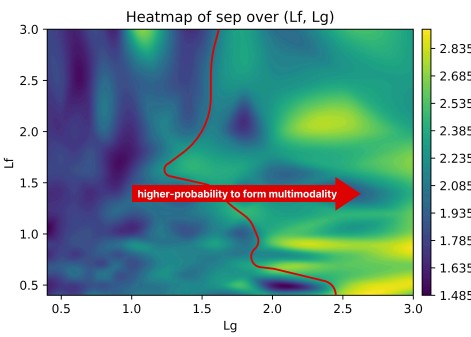

Figure 4: Heatmap of separated multimodality with varying Lipschitz constants.

threshold. This provides numerical evidence for the theorem: excessive contractivity limits the expressive power of NSDEs and prevents separated multimodality.

**Effect of Input Window Length.** We conducted an ablation study with $k \in \{50, 100, 150\}$, summarized in Figure 5. **MM-NSDE consistently achieves extremely low MISE** ($0.04 \to 0.017 \to 0.011$), demonstrating robustness to the historical window size. Nonetheless, a sufficiently large input window is required for the state-awareness module to fully capture latent regime transitions. In the main experiments, we set $k = 100$ as a practical trade-off between performance and efficiency.

**Scalability to High-Dimensional Sequences.** We design synthetic multivariate financial time series where the data-generating process switches between two configurations. Specifically, we simulate two correlated asset prices, $S_1(t)$ and $S_2(t)$, with state-drifts, volatilities, and changing correlation between the driving Brownian motions. This setup induces non-stationary dependencies and multimodal endpoint distributions, resembling realistic market fluctuations (full configuration

in Appendix 4). Figure 6 compares the ground-truth and model-generated endpoint distributions, showing that MM-NSDE accurately captures the joint dynamics of $S_1$ and $S_2$.

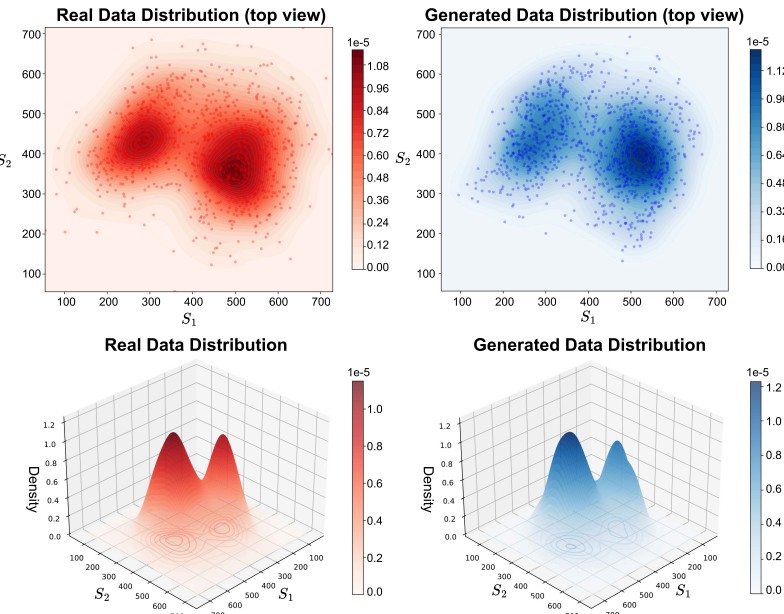

Figure 6: Comparison of real and generated endpoint distributions at $t = 252$ for the synthetic financial series. Left: ground-truth distribution of $S_1$ and $S_2$; Right: MM-NSDE results, accurately recovering the multimodal density induced by DGP switching.

In addition, we further validated the robustness of MM-NSDE on ultra high-dimensional data such as images. The detailed results can be found in the Appendix 4.

**Computational Efficiency Analysis.** In addition to the theoretical analysis discussed in Section 3.3, we evaluated alternative loss functions. As shown in Table 3, the results reveal trade-offs among accuracy, efficiency, and memory. MSE yields the largest error, and MMD, lacking entropic regularization, remains too rigid to capture complex distributional structures. Sinkhorn-based variants perform best overall: the baseline setting ($N = 512$, $\epsilon = 0.1$, iter=100) yields the highest accuracy at the cost of runtime and memory, whereas mini-batch and online kernel variants reduce resource demands with minimal loss in accuracy. By contrast, high blur severely harms performance, while low blur approaches optimal accuracy but incurs prohibitive computational overhead.

| Method (Loss) | Total Time [$10^3$ s] | Peak Memory [GB] | Final MISE |
|---|---|---|---|
| MSE | 1.21 | 15.24 | 0.108 |
| MMD (RBF, $\sigma = 0.5$) | 1.57 | 16.89 | 0.049 |
| Sinkhorn (Baseline, $N = 512$, $\epsilon = 0.1$, iter=100) | 1.98 | 32.55 | **0.004** |
| Sinkhorn w/ Mini-batch ($N = 128$, $\epsilon = 0.1$, iter=100) | **1.48** | 17.53 | 0.011 |
| Sinkhorn w/ Online Kernel ($N = 512$, $\epsilon = 0.1$, iter=100) | 2.00 | 18.16 | 0.009 |
| Sinkhorn w/ High Blur ($N = 512$, $\epsilon = 1.0$, iter=100) | 1.68 | 32.19 | 0.048 |
| Sinkhorn w/ Low Blur ($N = 512$, $\epsilon = 0.01$, iter=100) | 2.35 | 32.89 | 0.005 |
| Sinkhorn w/ Fewer Iter ($N = 512$, $\epsilon = 0.1$, iter=50) | 1.52 | 32.48 | 0.008 |
| Sinkhorn w/ Sparse Sampling ($N = 256$, $\epsilon = 0.1$, iter=100) | 1.72 | 24.32 | 0.006 |

Table 3: Performance comparison of different loss functions and Sinkhorn variants within the MM-NSDE framework. **Mini-batch** means computing Sinkhorn loss on randomly subsampled batches ($N = 128$) to reduce complexity. **Online Kernel** replaces the full $N \times N$ cost matrix with a streaming kernelized approximation to save memory. **Blur** refers to the entropic regularization coefficient $\epsilon$, where a larger $\epsilon$ produces smoother (blurred) transport plans. **Iter** denotes the number of Sinkhorn iterations used in optimization. **Sparse Sampling** reduces the number of support points ($N = 256$) for approximating the cost matrix. All experiments were conducted on an NVIDIA A800 GPU (80GB) using the GBM-1 dataset, with other experimental settings kept identical.

## ETHICS STATEMENT AND REPRODUCIBILITY STATEMENT

This paper aims to advance the field of Machine Learning. While the work may have potential societal implications, we do not identify any specific ethical concerns that require special attention. We provide sufficient details of the model, training procedure, and evaluation setup to allow independent reproduction of our results. All hyperparameters, datasets, and experimental settings are documented in the paper or supplementary material.

## REPRODUCIBILITY CHECKLIST

**Instructions for Authors:**

This document outlines key aspects for assessing reproducibility. Please provide your input by editing this `.tex` file directly.

For each question (that applies), replace the "Type your response here" text with your answer.

**Example:** If a question appears as

```
\question{Proofs of all novel claims are included}
{(yes/partial/no)}
Type your response here
```

you would change it to:

```
\question{Proofs of all novel claims are included}
{(yes/partial/no)}
yes
```

Please make sure to:

- Replace ONLY the "Type your response here" text and nothing else.

- Use one of the options listed for that question (e.g., **yes**, **no**, **partial**, or **NA**).

- **Not** modify any other part of the `\question` command or any other lines in this document.

You can `\input` this .tex file right before `\end{document}` of your main file or compile it as a stand-alone document. Check the instructions on your conference's website to see if you will be asked to provide this checklist with your paper or separately.

### 1. General Paper Structure

1.1. Includes a conceptual outline and/or pseudocode description of AI methods introduced (yes/partial/no/NA) yes

1.2. Clearly delineates statements that are opinions, hypothesis, and speculation from objective facts and results (yes/no) yes

1.3. Provides well-marked pedagogical references for less-familiar readers to gain background necessary to replicate the paper (yes/no) yes

### 2. Theoretical Contributions

2.1. Does this paper make theoretical contributions? (yes/no) yes

 If yes, please address the following points:

2.2. All assumptions and restrictions are stated clearly and formally (yes/partial/no) yes

2.3. All novel claims are stated formally (e.g., in theorem statements) (yes/partial/no) yes

2.4. Proofs of all novel claims are included (yes/partial/no) yes

2.5. Proof sketches or intuitions are given for complex and/or novel results (yes/partial/no) yes

2.6. Appropriate citations to theoretical tools used are given (yes/partial/no) yes

2.7. All theoretical claims are demonstrated empirically to hold (yes/partial/no/NA) yes

2.8. All experimental code used to eliminate or disprove claims is included (yes/no/NA) yes

## 3. Dataset Usage

3.1. Does this paper rely on one or more datasets? (yes/no) Type your response here

If yes, please address the following points:

3.2. A motivation is given for why the experiments are conducted on the selected datasets (yes/partial/no/NA) yes

3.3. All novel datasets introduced in this paper are included in a data appendix (yes/partial/no/NA) yes

3.4. All novel datasets introduced in this paper will be made publicly available upon publication of the paper with a license that allows free usage for research purposes (yes/partial/no/NA) yes

3.5. All datasets drawn from the existing literature (potentially including authors' own previously published work) are accompanied by appropriate citations (yes/no/NA) yes

3.6. All datasets drawn from the existing literature (potentially including authors' own previously published work) are publicly available (yes/partial/no/NA) yes

3.7. All datasets that are not publicly available are described in detail, with explanation why publicly available alternatives are not scientifically satisficing (yes/partial/no/NA) NA

## 4. Computational Experiments

4.1. Does this paper include computational experiments? (yes/no) yes

If yes, please address the following points:

4.2. This paper states the number and range of values tried per (hyper-) parameter during development of the paper, along with the criterion used for selecting the final parameter setting (yes/partial/no/NA) yes

4.3. Any code required for pre-processing data is included in the appendix (yes/partial/no) yes

4.4. All source code required for conducting and analyzing the experiments is included in a code appendix (yes/partial/no) yes

4.5. All source code required for conducting and analyzing the experiments will be made publicly available upon publication of the paper with a license that allows free usage for research purposes (yes/partial/no) yes

4.6. All source code implementing new methods have comments detailing the implementation, with references to the paper where each step comes from (yes/partial/no) yes

4.7. If an algorithm depends on randomness, then the method used for setting seeds is described in a way sufficient to allow replication of results (yes/partial/no/NA) yes

4.8. This paper specifies the computing infrastructure used for running experiments (hardware and software), including GPU/CPU models; amount of memory; operating system; names and versions of relevant software libraries and frameworks (yes/partial/no) no

4.9. This paper formally describes evaluation metrics used and explains the motivation for choosing these metrics (yes/partial/no) yes

4.10. This paper states the number of algorithm runs used to compute each reported result (yes/no) yes

4.11. Analysis of experiments goes beyond single-dimensional summaries of performance (e.g., average; median) to include measures of variation, confidence, or other distributional information (yes/no) yes

4.12. The significance of any improvement or decrease in performance is judged using appropriate statistical tests (e.g., Wilcoxon signed-rank) (yes/partial/no) no

4.13. This paper lists all final (hyper-)parameters used for each model/algorithm in the paper's experiments (yes/partial/no/NA) yes

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

CONTENTS

## DETAILS OF EXPERIMENTAL SETUP

### DATASETS

**Simulated Data.** Table 4 illustrates the first benchmark category, intra-family switching, where transitions remain within the same SDE family. Table 5 further details the parameter settings used for both intra-family and inter-family switching. For each setting, we generate 10,000 trajectories, with 2,000 held out for evaluation. In all cases, the task is formulated as conditional prediction: the model observes the first 100 steps and forecasts the following 50.

| Model Type | Data Generation Process | SDE1 Parameters | SDE2 Parameters | SDE3 Parameters |
|---|---|---|---|---|
| GBM | $dx_t = \begin{cases} \mu_1 x_t dt + \sigma_1 x_t dW_t, & t < 100 \\ \mu_1 x_t dt + \sigma_1 x_t dW_t, & t \geq 100, \text{probability } p \\ \mu_2 x_t dt + \sigma_2 x_t dW_t, & t \geq 100, \text{probability } 1-p \end{cases}$ | $\mu_1 = -4.0/365, \sigma_1 = 0.01$ $\mu_2 = 3.0/365, \sigma_2 = 0.01$ $p = 0.8$ | $\mu_1 = -4.0/365, \sigma_1 = 0.004$ $\mu_2 = -1.0/365, \sigma_2 = 0.006$ $p = 0.2$ | $\mu_1 = -1.0/365, \sigma_1 = 0.004$ $\mu_2 = 4.0/365, \sigma_2 = 0.006$ $p = 0.2$ |
| OU | $dx_t = \begin{cases} \theta_1(\mu_1 - x_t)dt + \sigma_1 dW_t, & t < 100 \\ \theta_1(\mu_1 - x_t)dt + \sigma_1 dW_t, & t \geq 100, \text{probability } p \\ \theta_2(\mu_2 - x_t)dt + \sigma_2 dW_t, & t \geq 100, \text{probability } 1-p \end{cases}$ | $\theta_1 = 0.01, \mu_1 = 50.0, \sigma_1 = 0.1$ $\theta_2 = 0.02, \mu_2 = 35.0, \sigma_2 = 0.2$ $p = 0.5$ | $\theta_1 = 0.05, \mu_1 = 30.0, \sigma_1 = 0.1$ $\theta_2 = 0.02, \mu_2 = 45.0, \sigma_2 = 0.3$ $p = 0.5$ | $\theta_1 = 0.02, \mu_1 = 35.0, \sigma_1 = 0.1$ $\theta_2 = 0.05, \mu_2 = 25.0, \sigma_2 = 0.3$ $p = 0.5$ |
| CIR | $dx_t = \begin{cases} \kappa_1(\theta_1 - x_t)dt + \sigma_1\sqrt{x_t}dW_t, & t < 100 \\ \kappa_1(\theta_1 - x_t)dt + \sigma_1\sqrt{x_t}dW_t, & t \geq 100, \text{probability } p \\ \kappa_2(\theta_2 - x_t)dt + \sigma_2\sqrt{x_t}dW_t, & t \geq 100, \text{probability } 1-p \end{cases}$ | $\kappa_1 = 0.02, \theta_1 = 50.0, \sigma_1 = 0.04$ $\kappa_2 = 0.05, \theta_2 = 40.0, \sigma_2 = 0.03$ $p = 0.5$ | $\kappa_1 = 0.02, \theta_1 = 30.0, \sigma_1 = 0.04$ $\kappa_2 = 0.05, \theta_2 = 40.0, \sigma_2 = 0.05$ $p = 0.5$ | $\kappa_1 = 0.02, \theta_1 = 30.0, \sigma_1 = 0.05$ $\kappa_2 = 0.05, \theta_2 = 20.0, \sigma_2 = 0.04$ $p = 0.5$ |

Table 4: **Parameter configurations for intra-family switching.** Transitions within the same SDE type are shown for SDE1–SDE3 across GBM, OU, and CIR.

For multidimensional data, we consider a two-asset stochastic system where the log-prices follow correlated Brownian motions with time-varying correlation. Specifically, let $W_1(t)$ and $W_2(t)$ denote Brownian motions with instantaneous correlation $\rho(t)$. The log-price dynamics are governed by the coupled stochastic differential equations:

$$dS_1(t) = \mu_1(t)S_1(t)\,dt + \sigma_1(t)S_1(t)\,dW_1(t),$$
$$dS_2(t) = \mu_2(t)S_2(t)\,dt + \sigma_2(t)S_2(t)\,dW_2(t).$$

with $\text{Corr}(dW_1(t), dW_2(t)) = \rho(t)$. The drift–volatility parameters $\{\mu_1, \mu_2, \sigma_1, \sigma_2, \rho\}$ evolve according to a two-regime switching mechanism: **Baseline regime:** $\mu_1 = 0.1$, $\mu_2 = 0.05$, $\sigma_1 = 0.2$, $\sigma_2 = 0.3$, $\rho = 0.5$. **Switched regime:** $\mu_1 = 0.05$, $\mu_2 = 0.1$, $\sigma_1 = 0.25$, $\sigma_2 = 0.15$, $\rho = -0.2$. At each discrete timestep, the system transitions from the baseline to the switched regime with probability $p_{\text{switch}} = 0.3$, mimicking abrupt market shifts. The initial conditions are set as $S_1(0) = S_2(0) = 100$. The discretization step is fixed at $\Delta t = 1/252$, corresponding to daily increments under a yearly horizon.

**Real-world Datasets.** We evaluate our method on three domains: **(i) Stock**, **(ii) Cryptocurrency**, and **(iii) Environment**. The stock domain includes time-series price data for AAL (American Airlines Group), ADBE (Adobe Inc.), and ADM (Archer Daniels Midland Company), spanning September 11, 2017 to February 16, 2018, with 5-minute sampling intervals. The cryptocurrency domain consists of Bitcoin high-price data at 1-minute intervals from December 29, 2024 to January 13, 2025, providing a high-frequency view of market dynamics. The environmental domain contains waste management records from Boralasgamuwa, Homagama, and Ballarat, merged into a unified dataset with 9,608 entries spanning July 3, 2000 to December 31, 2018. All datasets are divided into training and testing sets with an 8:2 ratio.

### ALGORITHM IMPLEMENT DETAILS

In the final experimental setup, we ensured that each baseline was configured with model-specific yet comparable hyperparameters. Autoformer used 2 encoder and 2 decoder layers with hidden size 256 and dropout 0.1. TimesNet employed 4 temporal convolutional blocks with hidden size 256 and SiLU activations. NS-Transformer was configured with 2 encoder layers, hidden size 128, and 4 attention heads. SegRNN contained 2 layers with hidden size 256, while Mamba used 2 selective-scan layers with hidden size 256. For stochastic baselines, both Latent-SDE and GAN-SDE parameterized the drift and diffusion terms with two-layer MLPs (128 hidden units, ReLU activations) under the same SDE solver and discretization scheme. Within MM-NSDE, the drift and diffusion functions were also modeled by two-layer MLPs with hidden size 256 and ReLU activations. The state-aware

module was implemented with stacked two blocks consisting of input projection, 1D convolution and SiLU activation. The parameter choices were determined through systematic grid search: we uniformly searched learning rates $\{1e\text{-}3, 5e\text{-}4, 1e\text{-}4\}$, hidden sizes $\{64, 128, 256\}$, number of layers $\{2, 4, 6\}$, dropout $\{0.1, 0.3, 0.5\}$, and weight decay $\{0, 1e\text{-}4, 1e\text{-}3\}$, selecting the best-performing configurations on the validation set. For stochastic models, we additionally tuned SDE discretization and noise intensity parameters. All experiments were repeated with five random seeds, and we report the mean preformance, ensuring robustness and fairness in comparisons across models.

### EVALUATION METRICS

We employ four complementary metrics to evaluate model performance. First, MISE assesses the discrepancy between true and generated distributions under hybrid conditioning paths, defined as

$$\text{MISE} = \mathbb{E}_{y_{1:L+T} \sim p_{\text{true}}} \left[ \mathbb{E}_{\hat{y}_{L+1:L+T} \sim p_{\text{model}}(\cdot | y_{1:L})} \left[ \int_{-\infty}^{\infty} \left( \hat{M}_T(y_{L+T} \mid \mathcal{H}_{\text{h}}) - M_T(y_{L+T} \mid \mathcal{H}_{\text{t}}) \right)^2 dy_{L+T} \right] \right],$$

where $\mathcal{H}_{\text{h}} = \{\hat{y}_{1:L}, \hat{y}_{L+1:L+T}\}$ combines true history and model predictions, and $\mathcal{H}_{\text{t}} = y_{1:L+T}$ denotes the complete true path. Second, for extreme risk assessment, we use the TD metric, which measures discrepancies in the lower and upper $5\%$ quantiles of the cumulative distribution function (CDF):

$$\Delta_{\text{T}} = \int_{-\infty}^{F^{-1}(\alpha)} \left| F(x) - \hat{F}(x) \right| dx + \int_{F^{-1}(1-\alpha)}^{\infty} \left| F(x) - \hat{F}(x) \right| dx,$$

where $F(x)$ and $\hat{F}(x)$ denote the true and empirical CDFs, respectively, with $\alpha = 0.05$. Third, we compute the MMD to quantify the overall distributional gap between true and generated samples:

$$\text{MMD}^2(\mathcal{H}_{\text{t}}, \mathcal{H}_{\text{h}}) = \mathbb{E}_{x,x' \sim p_{\text{true}}}[k(x, x')] + \mathbb{E}_{y,y' \sim p_{\text{model}}}[k(y, y')] - 2\,\mathbb{E}_{x \sim p_{\text{true}}, y \sim p_{\text{model}}}[k(x, y)],$$

where $k(\cdot, \cdot)$ is a positive definite kernel (RBF kernel in our experiments). Finally, we include the MSE to measure pointwise prediction accuracy:

$$\text{MSE} = \frac{1}{T} \sum_{t=1}^{T} \mathbb{E}\left[ (y_{L+t} - \hat{y}_{L+t})^2 \right],$$

which evaluates the expected squared error between true trajectories and model predictions under teacher forcing.

| Model Type | Data Generation Process | SDE Parameters |
|---|---|---|
| GBM $\rightarrow$ GBM or OU | $dx_t = \begin{cases} \mu_1 x_t dt + \sigma_1 x_t dW_t, & t < 100 \\ \mu_1 x_t dt + \sigma_1 x_t dW_t, & t \geq 100, p_1 \\ \theta_2(\mu_2 - x_t)dt + \sigma_2 dW_t, & t \geq 100, p_2 \end{cases}$ | $\mu_1 = -8.0/365, \sigma_1 = 0.04, \theta_2 = 0.02, \mu_2 = 30.0, \sigma_2 = 0.8, p_1 = 0.7, p_2 = 0.3$ |
| GBM $\rightarrow$ GBM or CIR | $dx_t = \begin{cases} \mu_1 x_t dt + \sigma_1 x_t dW_t, & t < 100 \\ \mu_1 x_t dt + \sigma_1 x_t dW_t, & t \geq 100, p_1 \\ \kappa_2(\theta_2 - x_t)dt + \sigma_2 \sqrt{x_t}dW_t, & t \geq 100, p_2 \end{cases}$ | $\mu_1 = -8.0/365, \sigma_1 = 0.04, \kappa_2 = 0.003, \theta_2 = 100.0, \sigma_2 = 0.20, p_1 = 0.35, p_2 = 0.65$ |
| GBM $\rightarrow$ OU or CIR | $dx_t = \begin{cases} \mu_1 x_t dt + \sigma_1 x_t dW_t, & t < 100 \\ \theta_2(\mu_2 - x_t)dt + \sigma_2 dW_t, & t \geq 100, p_1 \\ \kappa_3(\theta_3 - x_t)dt + \sigma_3 \sqrt{x_t}dW_t, & t \geq 100, p_2 \end{cases}$ | $\mu_1 = -2.0/365, \sigma_1 = 0.04, \theta_2 = 0.02, \mu_2 = 100.0, \sigma_2 = 2.0, \kappa_3 = 0.003, \theta_3 = 100.0, \sigma_3 = 0.20, p_1 = 0.35, p_2 = 0.65$ |
| OU $\rightarrow$ OU or GBM | $dx_t = \begin{cases} \theta_1(\mu_1 - x_t)dt + \sigma_1 dW_t, & t < 100 \\ \theta_1(\mu_1 - x_t)dt + \sigma_1 dW_t, & t \geq 100, p_1 \\ \mu_2 x_t dt + \sigma_2 x_t dW_t, & t \geq 100, p_2 \end{cases}$ | $\theta_1 = 0.002, \mu_1 = -300.0, \sigma_1 = 0.08, \mu_2 = -0.4/365, \sigma_2 = 0.049, p_1 = 0.7, p_2 = 0.3$ |
| OU $\rightarrow$ OU or CIR | $dx_t = \begin{cases} \theta_1(\mu_1 - x_t)dt + \sigma_1 dW_t, & t < 100 \\ \theta_1(\mu_1 - x_t)dt + \sigma_1 dW_t, & t \geq 100, p_1 \\ \kappa_2(\theta_2 - x_t)dt + \sigma_2 \sqrt{x_t}dW_t, & t \geq 100, p_2 \end{cases}$ | $\theta_1 = 0.02, \mu_1 = 30.0, \sigma_1 = 0.8, \kappa_2 = 0.1, \theta_2 = 0.50, \sigma_2 = 0.20, p_1 = 0.7, p_2 = 0.3$ |
| OU $\rightarrow$ GBM or CIR | $dx_t = \begin{cases} \theta_1(\mu_1 - x_t)dt + \sigma_1 dW_t, & t < 100 \\ \mu_2 x_t dt + \sigma_2 x_t dW_t, & t \geq 100, p_1 \\ \kappa_3(\theta_3 - x_t)dt + \sigma_3 \sqrt{x_t}dW_t, & t \geq 100, p_2 \end{cases}$ | $\theta_1 = 0.3, \mu_1 = 0.0, \sigma_1 = 0.05, \mu_2 = -7.3/365, \sigma_2 = 0.05, \kappa_3 = 0.3, \theta_3 = 0.10, \sigma_3 = 0.05, p_1 = 0.4, p_2 = 0.6$ |
| CIR $\rightarrow$ OU or GBM | $dx_t = \begin{cases} \kappa_1(\theta_1 - x_t)dt + \sigma_1 \sqrt{x_t}dW_t, & t < 100 \\ \theta_2(\mu_2 - x_t)dt + \sigma_2 dW_t, & t \geq 100, p_1 \\ \mu_3 x_t dt + \sigma_3 x_t dW_t, & t \geq 100, p_2 \end{cases}$ | $\kappa_1 = 0.003, \theta_1 = 100.0, \sigma_1 = 0.20, \theta_2 = 0.02, \mu_2 = 100.0, \sigma_2 = 2.0, \mu_3 = -2.0/365, \sigma_3 = 0.04, p_1 = 0.4, p_2 = 0.6$ |
| CIR $\rightarrow$ CIR or OU | $dx_t = \begin{cases} \kappa_1(\theta_1 - x_t)dt + \sigma_1 \sqrt{x_t}dW_t, & t < 100 \\ \kappa_1(\theta_1 - x_t)dt + \sigma_1 \sqrt{x_t}dW_t, & t \geq 100, p_1 \\ \theta_2(\mu_2 - x_t)dt + \sigma_2 dW_t, & t \geq 100, p_2 \end{cases}$ | $\kappa_1 = 0.003, \theta_1 = 50.0, \sigma_1 = 0.20, \theta_2 = 0.02, \mu_2 = 100.0, \sigma_2 = 2.0, p_1 = 0.5, p_2 = 0.5$ |
| CIR $\rightarrow$ GBM or CIR | $dx_t = \begin{cases} \kappa_1(\theta_1 - x_t)dt + \sigma_1 \sqrt{x_t}dW_t, & t < 100 \\ \kappa_1(\theta_1 - x_t)dt + \sigma_1 \sqrt{x_t}dW_t, & t \geq 100, p_1 \\ \mu_2 x_t dt + \sigma_2 x_t dW_t, & t \geq 100, p_2 \end{cases}$ | $\kappa_1 = 0.01, \theta_1 = 70.0, \sigma_1 = 0.05, \mu_2 = -4.0/365, \sigma_2 = 0.04, p_1 = 0.6, p_2 = 0.4$ |

Table 5: SDE parameters configurations for switching between different model families.

## MORE ANALYSIS

### MORE RESULTS.

**Simulated Results.** As shown in Table 6 and Table 7, the results show that MM-NSDEs maintain competitive performance across nearly all evaluation metrics, achieving both high point-wise trajectory accuracy and distributional alignment. This is, to our knowledge, the first evidence that a model can perform well in both aspects without a trade-off, highlighting its capability of global generalization and local precision. Switching dynamics serve as the key stress test: while some baselines remain reasonable under intra-family scenarios, they fail in inter-family settings, with MSE rising to high levels. This indicates their inability to capture the mechanisms governing transitions across dynamical families. Latent-SDEs, for instance, work within a single dynamical regime but fail once family shifts occur. These findings suggest that not all SDE-based approaches are generalizable; the key lies in incorporating structured inductive biases that reflect the properties of real-world data to achieve cross-family generalization.

| Model | GBM2 | | | | GBM3 | | | | OU2 | | | |
|---|---|---|---|---|---|---|---|---|---|---|---|---|
| | MISE | TD | MMD | MSE | MISE | TD | MMD | MSE | MISE | TD | MMD | MSE |
| Autoformer | 0.16 | 0.08 | 0.17 | 2.06 | 0.32 | 0.09 | 0.08 | 1.66 | 1.30 | 0.04 | 0.09 | 1.81 |
| DLinear | 0.24 | 0.09 | 0.08 | 1.01 | 0.17 | 0.10 | 0.12 | 0.99 | 1.89 | 0.10 | 0.52 | 1.21 |
| Mamba | 0.13 | 0.07 | 0.15 | 2.01 | 0.30 | 0.09 | 0.07 | 1.62 | 1.36 | 0.04 | 0.09 | 1.73 |
| NS-Transformer | 0.10 | 0.09 | 0.05 | 1.13 | 0.10 | 0.10 | 0.06 | 1.18 | 1.29 | 0.05 | 0.09 | 1.75 |
| SegRNN | 0.12 | 0.07 | 0.14 | 2.01 | 0.28 | 0.09 | 0.07 | 1.61 | 1.42 | 0.05 | 0.09 | 1.73 |
| TimesNet | 0.12 | 0.06 | 0.15 | 2.10 | 0.31 | 0.09 | 0.06 | 1.59 | 1.37 | 0.04 | 0.09 | 1.74 |
| Latent-SDEs | 1.03 | 0.10 | 0.30 | 2.01 | 0.68 | 0.10 | 0.25 | 1.81 | 1.17 | 0.10 | 0.41 | 2.20 |
| GAN-SDEs | 3.66 | 0.10 | 0.51 | 3.00 | 0.54 | 0.10 | 0.30 | 2.00 | 1.12 | 0.10 | 0.45 | 2.50 |
| MM-NSDEs | **0.01** | **0.01** | **0.02** | **0.75** | **0.02** | 0.10 | **0.03** | **0.08** | **0.03** | **0.02** | **0.03** | **0.12** |

| Model | OU3 | | | | CIR2 | | | | CIR3 | | | |
|---|---|---|---|---|---|---|---|---|---|---|---|---|
| | MISE | TD | MMD | MSE | MISE | TD | MMD | MSE | MISE | TD | MMD | MSE |
| Autoformer | 0.74 | 0.01 | 0.17 | 1.47 | 0.44 | 0.01 | 0.09 | 1.80 | 0.63 | 0.02 | 0.11 | 1.61 |
| DLinear | 1.90 | 0.10 | 0.40 | 1.18 | 1.16 | 0.10 | 0.46 | 1.16 | 1.61 | 0.09 | 0.39 | 1.19 |
| Mamba | 0.76 | 0.03 | 0.19 | 1.41 | 0.42 | 0.00 | 0.12 | 1.53 | 0.60 | 0.02 | 0.11 | 1.58 |
| NS-Transformer | 0.75 | 0.02 | 0.17 | 1.47 | 0.41 | 0.03 | 0.20 | 1.87 | 0.17 | 0.04 | 0.10 | 1.65 |
| SegRNN | 0.72 | **0.01** | 0.21 | 1.41 | 0.45 | 0.02 | 0.13 | 1.49 | 0.63 | 0.01 | 0.12 | 1.57 |
| TimesNet | 0.76 | 0.02 | 0.19 | 1.41 | 0.44 | 0.01 | 0.13 | 1.51 | 0.65 | 0.02 | 0.11 | 1.57 |
| Latent-SDEs | 0.69 | 0.10 | 0.35 | 2.00 | 0.82 | 0.10 | 0.30 | 1.91 | 0.47 | 0.02 | 0.20 | 1.50 |
| GAN-SDEs | 0.59 | 0.10 | 0.41 | 2.31 | 0.92 | 0.10 | 0.35 | 2.20 | 0.71 | 0.10 | 0.26 | 1.80 |
| MM-NSDEs | **0.45** | 0.03 | **0.03** | **0.03** | **0.33** | **0.00** | **0.01** | **0.01** | **0.03** | 0.07 | **0.05** | **0.07** |

Table 6: Comparison of different models on the simulated dataset with Intra-family switching. Our results are highlighted with darker shading, and the best performance is shown in bold. The reported values are scaled by $10^{-2}$.

**Real-world Tasks.** We extend our evaluation to a public benchmark on uncertainty estimation in financial time series, a setting where modeling stochastic variability is essential for tasks such as risk management and portfolio optimization. As shown in Table 8, MM-NSDE reduces the MMD score by several orders of magnitude compared to all baselines. This demonstrates its ability to capture distributional uncertainty in financial dynamics, which is critical for stress testing and volatility forecasting.

### MM-NSDE FOR NON-SEQUENTIAL DATA MODELING

Differential equation (DE) based models can be viewed as a block, as prior work has shown the equivalence between residual networks and SDE formulations (Tzen and Raginsky, 2019; Kong et al., 2020). We benchmarked DE- and SDE-based blocks on the MNIST OOD detection task using three metrics: TNR@TPR95%, Detection Accuracy, and AUPR Out. As shown in Table 9, DE methods perform well on in-distribution data but achieve lower AUPR Out, indicating weaker rejection of OOD samples. SDE-based approaches provide stronger uncertainty estimation. Our MM-NSDE block achieves the best TNR@TPR95% and AUPR Out while maintaining competitive

| Model | OU → OU or GBM | | | | OU → OU or CIR | | | | OU → GBM or CIR | | | |
|---|---|---|---|---|---|---|---|---|---|---|---|---|
| | MISE | TD | MMD | MSE | MISE | TD | MMD | MSE | MISE | TD | MMD | MSE |
| Autoformer | 0.67 | 0.09 | 0.16 | 1.90 | 2.83 | 0.07 | 0.20 | 2.01 | 1.45 | 0.06 | 0.19 | 1.80 |
| DLinear | 0.24 | 0.01 | 0.11 | 1.26 | 2.81 | 0.06 | 0.29 | _1.62_ | 8.30 | 0.06 | 0.28 | _1.52_ |
| Mamba | 0.68 | 0.09 | 0.18 | 2.31 | 2.85 | _0.06_ | 0.14 | 1.74 | 1.44 | _0.06_ | _0.13_ | 1.62 |
| NS-Transformer | _0.23_ | _0.01_ | _0.11_ | _1.22_ | 2.85 | _0.02_ | 0.16 | 1.71 | 0.90 | _0.04_ | 0.14 | 1.60 |
| SegRNN | 0.67 | 0.10 | 0.19 | 2.36 | 2.91 | 0.07 | 0.15 | 1.92 | 1.48 | 0.06 | 0.13 | 1.74 |
| TimesNet | 0.67 | 0.09 | 0.19 | 2.34 | _2.84_ | 0.06 | _0.14_ | 1.83 | _1.44_ | 0.06 | 0.13 | 1.67 |
| Latent-SDEs | 0.66 | 0.10 | 0.35 | 1.81 | _1.00_ | 0.10 | 0.41 | 2.10 | 0.86 | 0.10 | 0.38 | 1.91 |
| GAN-SDEs | 1.40 | 0.10 | 0.46 | 2.30 | 1.40 | 0.10 | 0.50 | 2.61 | _0.63_ | 0.10 | 0.49 | 2.40 |
| MM-NSDEs | **0.07** | **0.07** | **0.05** | **0.06** | **0.09** | **0.02** | **0.01** | **0.04** | **0.12** | **0.19** | **0.01** | **0.00** |

| Model | CIR → CIR or GBM | | | | CIR → CIR or OU | | | | CIR → GBM or OU | | | |
|---|---|---|---|---|---|---|---|---|---|---|---|---|
| | MISE | TD | MMD | MSE | MISE | TD | MMD | MSE | MISE | TD | MMD | MSE |
| Autoformer | 0.44 | 0.09 | 0.13 | 2.24 | 0.08 | 0.03 | 0.04 | 1.20 | 0.73 | 0.05 | 0.03 | 1.16 |
| DLinear | 0.70 | 0.09 | 0.12 | 1.19 | 2.23 | 0.07 | 0.37 | 0.94 | 1.78 | 0.08 | 0.33 | 0.92 |
| Mamba | 0.37 | 0.07 | 0.09 | 1.97 | 0.10 | 0.02 | 0.05 | 1.07 | 0.79 | 0.04 | 0.04 | 1.02 |
| NS-Transformer | 0.26 | 0.09 | 0.10 | 1.50 | 0.08 | 0.01 | 0.05 | 1.01 | 0.27 | 0.03 | 0.08 | 1.11 |
| SegRNN | 0.39 | 0.08 | 0.11 | 2.11 | 0.08 | 0.01 | 0.04 | 1.08 | 0.72 | 0.03 | 0.03 | 1.00 |
| TimesNet | 0.40 | 0.07 | 0.10 | 2.05 | 0.09 | 0.01 | 0.05 | 1.09 | 0.72 | 0.04 | 0.03 | 1.03 |
| Latent-SDEs | 0.62 | 0.05 | 0.29 | 1.61 | 0.22 | 0.10 | 0.22 | 1.30 | 0.16 | 0.05 | 0.19 | 1.20 |
| GAN-SDEs | 2.60 | 0.10 | 0.38 | 2.21 | 1.80 | 0.10 | 0.34 | 1.80 | 0.41 | 0.10 | 0.28 | 1.71 |
| MM-NSDEs | **0.20** | **0.05** | **0.01** | **0.02** | **0.05** | **0.00** | **0.01** | **0.10** | **0.04** | **0.09** | **0.03** | **0.42** |

Table 7: Comparison of different models on the simulated dataset with Inter-family switching. Our results are highlighted with darker shading, and the best performance is shown in bold. The reported values are scaled by $10^{-2}$.

| Metric | DLinear | SegRNN | TimesNet | Autoformer | NS-Transformer | Mamba | Latent-SDEs | GAN-SDEs | **MM-NSDE** |
|---|---|---|---|---|---|---|---|---|---|
| MMD ($\times 10^{-3}$) ↓ | 240.76 | 210.89 | 205.12 | 195.33 | 98.61 | 215.47 | 455.18 | 510.42 | **0.28** |

Table 8: Performance on uncertainty estimation in financial time series (MMD ↓). Lower values indicate better distributional uncertainty modeling.

Detection Accuracy, showing that modeling network evolution as a stochastic process improves OOD detection compared to DE-based counterparts.

| Model | TNR@TPR95% | Detect. Acc. | AUPR Out |
|---|---|---|---|
| Threshold | $94.0 \pm 1.4$ | $94.8 \pm 0.7$ | $89.4 \pm 1.1$ |
| MC-Dropout | $92.9 \pm 1.6$ | $94.2 \pm 0.7$ | $88.5 \pm 1.7$ |
| PN | $93.4 \pm 2.2$ | $94.5 \pm 1.1$ | $88.5 \pm 1.3$ |
| BBP | $75.0 \pm 3.4$ | $90.4 \pm 2.2$ | $76.0 \pm 4.2$ |
| p-SGLD | $85.3 \pm 2.3$ | $90.5 \pm 1.3$ | $82.8 \pm 2.2$ |
| SDE-Net | $99.6 \pm 0.2$ | $98.6 \pm 0.5$ | $99.5 \pm 0.3$ |
| MM-NSDE | **99.7** $\pm 0.2$ | _98.3_ $\pm 0.4$ | **99.9** $\pm 0.1$ |

Table 9: Full comparison of all baselines on MNIST OOD detection using three representative metrics. Best results are in **bold**, and second best are underlined.

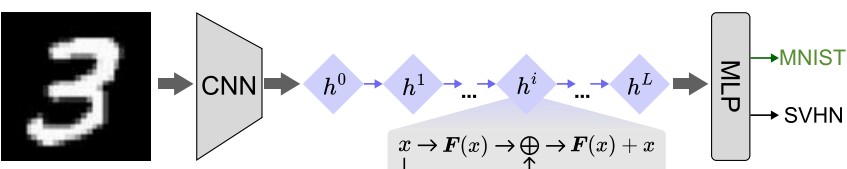

Figure 7: Illustration of the MM-NSDE block applied to non-sequential data (image classification). Each residual connection can be interpreted as a discretization step $dt$ of an underlying stochastic differential equation, allowing the network depth to be viewed as the time axis.

MODEL SCALABILITY ANALYSIS

Figure 8 illustrates the performance of different models as their parameter scales increase. As the number of parameters grows, Mamba shows a steady decrease in error, indicating good scalability. SegRNN performs relatively well at smaller scales, but its error initially increases as the model size grows, before improving again at larger scales. This pattern may reflect differences in optimization and generalization behaviors across capacity ranges. Autoformer maintains its error within a relatively stable range across the examined scales. In contrast, MM-NSDE consistently achieves error levels that are substantially lower than all other models, and its performance further improves with scale, highlighting its strong scalability advantage.

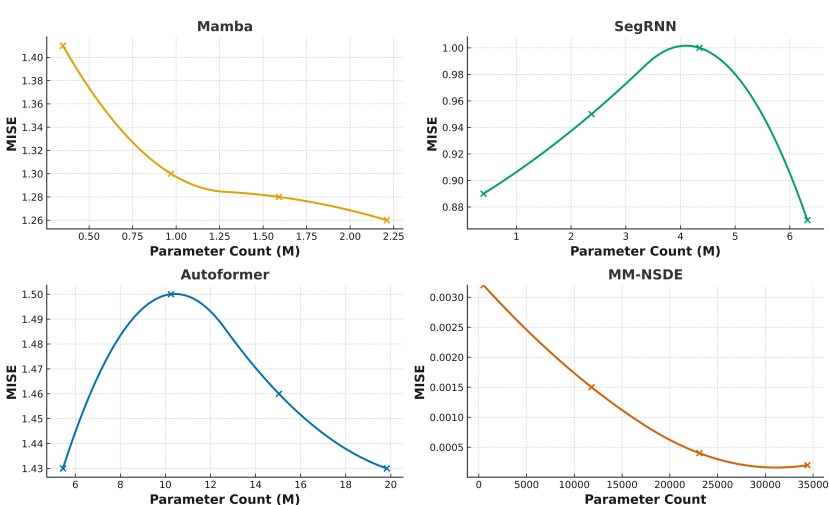

Figure 8: Comparison of model performance across different parameter scales on the GBM1 simulated dataset, with all other experimental settings kept consistent with the previous experiments.

ANOTHER PERSPECTIVE ON THE FAILURE OF NSDES

Another perspective on why NSDEs struggle to capture multimodal behaviors lies in their *structural limitations*. Although the drift and diffusion terms are parameterized by expressive neural networks, the underlying SDE dynamics are still continuous stochastic flows. Such flows inherently preserve smoothness and local regularity, which naturally bias the resulting transition densities toward unimodality. Consequently, multimodality can only arise under restrictive conditions, and these are rarely satisfied in practice.

For instance, the GBM, widely applied in finance and economics, produces a log-normal transition density that is strictly unimodal. More generally, many classical SDEs share this unimodal bias. To formalize when multimodality is possible, we provide the following theorem:

**Theorem 5.** *Suppose $Y_t$ follows the NSDE in Eq. 1. Suppose $f$ and $g$ are Lipschitz continuous functions, and $f(Y_t, t)$ and $g(Y_t, t)$ have finite moments. If Assumptions 11–12 in Appendix A hold, then the stationary transition density is multimodal.*

Theorem 5 establishes a theoretical framework for determining when multimodality can occur. However, these conditions are typically violated by standard SDEs. As a concrete example, the following corollary shows that the Ornstein–Uhlenbeck (OU) process always yields a unimodal stationary distribution.

**Corollary 6** (Ornstein–Uhlenbeck process). *Suppose $Y_t$ follows the Ornstein–Uhlenbeck process*

$$dY_t = \kappa(\mu - Y_t)dt + \sigma dW_t, \quad \kappa > 0, \ \sigma > 0,$$

*then the stationary transition density of $Y_t$, $p(Y_t|Y_{t-1})$, is unimodal.*

*Proof.* We provide only a brief outline here; the full derivation can be found in Appendix 4. By Theorem 5, the stationary point is $Y^* = \mu$. At this point, $\gamma(Y^*) = -\kappa/\sigma^2 < 0$[1]. Hence, the stationary transition density of the Ornstein–Uhlenbeck process is strictly unimodal. □

---

[1]$Y^*$ and $\gamma(\cdot)$ are defined in Assumptions 11–12

PROOFS

PROOF FOR THEOREM 3

We begin by stating the assumptions used throughout.

**Assumption 7** (Coefficient regularity). *The drift and diffusion satisfy: 1) global Lipschitz continuity in $y$, with constants $L_f, L_g$; 2) uniform boundedness: $\|g\|_\infty := \sup_{(y,s)} \|g_{\theta_2}(y,s)\|_{\mathrm{op}} < \infty$; 3) uniform ellipticity: $g(y,s)g(y,s)^\top \succeq \alpha I_d$ for some $\alpha > 0$.*

**Assumption 8** (Separated bimodality). *The terminal law $\nu = \mathcal{L}(Y_t)$ has separated bimodality along some $u \in \mathbb{S}^{d-1}$. Specifically, the one-dimensional marginal density $q_u$ satisfies*

$$q_u(s) \ \leq \ \lambda \frac{1}{\sigma\sqrt{2\pi}} e^{-\frac{(s-m_1)^2}{2\sigma^2}} + (1-\lambda)\frac{1}{\sigma\sqrt{2\pi}} e^{-\frac{(s-m_2)^2}{2\sigma^2}},$$

*for some $m_1 < m_2$, separation $\delta := m_2 - m_1 > 0$, scale $\sigma > 0$, and mixture weight $\lambda \in (0,1)$.*

**Proposition 9** (Content bound for separated mixtures). *Let $\nu$ be a probability measure on $\mathbb{R}^d$ whose projection onto some $u \in \mathbb{S}^{d-1}$ has density $q_u$ satisfying the separated bimodality condition in Assumption 8 with means $m_1 < m_2$, separation $\delta = m_2 - m_1 > 0$, and variance parameter $\sigma^2$. Then the half-space profile $J_\nu(\lambda)$ satisfies*

$$J_\nu(\lambda) \ \leq \ \frac{1}{\sigma\sqrt{2\pi}} \exp\Big(-\frac{\delta^2}{8\sigma^2}\Big), \qquad \forall \lambda \in (0,1).$$

*Proof.* By the isoperimetric transport inequality , if $T$ is $L$-Lipschitz and $T_\#\mu = \nu$, then

$$L \ \geq \ \sup_{\lambda \in (0,1)} \frac{I(\lambda)}{J_\nu(\lambda)},$$

where $I(\lambda) = \varphi(\Phi^{-1}(\lambda))$ is the Gaussian half-space profile and $J_\nu(\lambda)$ is the half-space profile of $\nu$. Under Assumption 8, Proposition 9 ensures

$$J_\nu(\lambda) \ \leq \ \frac{1}{\sigma\sqrt{2\pi}} \exp\Big(-\frac{\delta^2}{8\sigma^2}\Big),$$

so that

$$L \ \geq \ \frac{\varphi(\Phi^{-1}(\lambda))}{J_\nu(\lambda)} \ \geq \ \sigma \exp\Big(\frac{\delta^2}{8\sigma^2} - \tfrac{1}{2}(\Phi^{-1}(\lambda))^2\Big).$$

On the other hand, standard variational estimates for SDEs with globally Lipschitz coefficients (see, e.g., Friz–Victoir) yield that the Itô–Lyons map $\Gamma$ of the NSDE is Lipschitz with

$$L_{\mathrm{NSDE}} \ \leq \ c_0 \|g\|_\infty \exp\big(c_1(L_f + L_g^2)t\big),$$

where $c_0, c_1$ depend only on the dimension and ellipticity. Since both bounds apply to the same map $\Gamma$, we conclude

$$c_0 \|g\|_\infty \exp\big(c_1(L_f + L_g^2)t\big) \ \geq \ \sigma \exp\Big(\frac{\delta^2}{8\sigma^2} - \tfrac{1}{2}(\Phi^{-1}(\lambda))^2\Big),$$

which is the claimed inequality. $\qquad\square$

PROOF OF PROPOSITION 4

**Assumption 10** (Separated multimodal distributions). *Let*

$$p = \tfrac{1}{2}\delta_{-a/2} + \tfrac{1}{2}\delta_{a/2}, \quad q = \tfrac{1}{2}\delta_{-a/2} + \tfrac{1}{2}\delta_{a/2+1},$$

*with mode separation $a \to \infty$. Generalizations to higher dimensions or Gaussian mixtures follow analogously.*

*Proof.* (1) Wasserstein-2:

$$W_2^2(p, q) \ge \tfrac{1}{2} \cdot a^2 \quad \Rightarrow \quad W_2(p, q) = \Omega(a).$$

(2) Sinkhorn ($\varepsilon$-regularized OT):

$$S_\varepsilon(p, q) \ge W_2(p, q) - C(\varepsilon),$$

hence $S_\varepsilon(p, q) = \Omega(a)$.

(3) MMD with Gaussian kernel $k(x, y) = \exp(-\|x - y\|^2/\sigma^2)$:

$$\mathrm{MMD}^2(p, q) = \mathbb{E}_p[k(x, x')] + \mathbb{E}_q[k(y, y')] - 2\mathbb{E}_{p,q}[k(x, y)].$$

As $a \to \infty$, cross-mode terms $\to 0$, leaving only bounded within-mode terms. Thus $\mathrm{MMD}(p, q) = O(1)$. $\qquad\square$

PROOF OF THEOREM 5

**Assumption 11.** *Equation* $f(Y_t, t) - g(Y_t, t)\frac{\partial}{\partial Y}g(Y_t, t) = 0$ *has at least one solution at* $Y^*$

**Assumption 12.** *Define* $\gamma(Y_t, t) = \frac{f'(Y_t,t)g^2(Y_t,t) - (g'(Y_t,t))^2 g^2(Y_t,t) - 2f(Y_t,t)g'(Y_t,t) - g(Y_t,t)(g'(Y_t,t))^2}{g^4(Y_t,t)}$.
*At* $Y^*$, $\gamma(Y^*, t)$ *is strictly positive.*

*Proof.* The Fokker–Planck equation for the transition density $p(Y_t|Y_{t-1}, t)$ is given by:

$$\frac{\partial}{\partial t}p(Y_t|Y_{t-1}, t) = -\frac{\partial}{\partial Y}[f(Y_t, t)p(Y_t|Y_{t-1}, t)] + \frac{\partial^2}{\partial Y^2}[\frac{g^2(Y_t, t)}{2}p(Y_t|Y_{t-1}, t)]$$

Suppose that the transition density is stationary, that is, $\frac{\partial}{\partial t}p(Y_t|Y_{t-1}, t) = 0$, then the Fokker–Planck equation degenerates to

$$-\frac{\partial}{\partial Y}[f(Y_t, t)p(Y_t|Y_{t-1})] + \frac{\partial^2}{\partial Y^2}[\frac{g^2(Y_t, t)}{2}p(Y_t|Y_{t-1})] = 0$$

where $p(Y_t|Y_{t-1}) = \lim_{t \to \infty} p(Y_t|Y_{t-1}, t)$. Then, there must be a constant $C$ s.t.

$$\frac{\partial}{\partial Y}\frac{g^2(Y_t, t)}{2} * p(Y_t|Y_{t-1}) - f(Y_t, t)p(Y_t|Y_{t-1}) = C.$$

We take an integral for both sides of the above equation on the real line and obtain:

$$\left|\int_{-\infty}^{\infty} C dY_t\right| = \left|\int_{-\infty}^{\infty}(\frac{\partial}{\partial Y}\frac{g^2(Y_t, t)}{2}p(Y_t|Y_{t-1}) - f(Y_t, t)p(Y_t|Y_{t-1}))dY_t\right|$$

$$\le \left|\int_{-\infty}^{\infty} p(Y_t|Y_{t-1})g(Y_t, t)\frac{\partial}{\partial Y}g(Y_t, t)dY_t\right| + \left|\int_{-\infty}^{\infty}\frac{g^2(Y_t, t)}{2}\frac{\partial}{\partial Y}p(Y_t|Y_{t-1})dY_t\right|$$

$$+ \left|\int_{-\infty}^{\infty} p(Y_t|Y_{t-1})f(Y_t, t)dY_t\right|$$

$$< \infty$$

Therefore, $C$ must be zero. This gives us a differential equation for the stationary transitional density:

$$\frac{\partial}{\partial Y}p(Y_t|Y_{t-1}) = \frac{2}{g^2(Y_t, t)}(f(Y_t, t) - g(Y_t, t)\frac{\partial}{\partial Y}g(Y_t, t))p(Y_t|Y_{t-1}).$$

It is easy to verify that if Assumption 11 is satisfied, then $\exists\, Y^*$ s.t. $\frac{\partial}{\partial Y}p(Y^*) = 0$. Further, because

$$\frac{\partial}{\partial Y}\frac{\partial p(Y_t|Y_{t-1})/\partial Y}{p(Y_t|Y_{t-1})} \propto \gamma(Y_t, t).$$

Under Assumption 12, we have

$$\frac{\partial^2}{\partial Y^2}p(Y^*) > 0.$$

This means the stationary transitional density has at least one minimum at $Y^*$, i.e., at least two local maximums. $\qquad\square$

PROOF OF COROLLARY 6

*Proof.* The Fokker–Planck equation for the transition density $p(Y_t|Y_{t-1}, t)$ is given by:

$$\frac{\partial}{\partial t}p(Y_t|Y_{t-1}, t) = -\frac{\partial}{\partial Y}[\kappa(\mu - Y_t)p(Y_t|Y_{t-1}, t)] + \frac{\partial^2}{\partial Y^2}[\frac{\sigma^2}{2}p(Y_t|Y_{t-1}, t)]$$

Suppose that the transition density is stationary, that is, $\frac{\partial}{\partial t}p(Y_t|Y_{t-1}, t) = 0$, then the Fokker–Planck equation degenerates to

$$-\frac{\partial}{\partial Y}[\kappa(\mu - Y_t)p(Y_t|Y_{t-1})] + \frac{\partial^2}{\partial Y^2}[\frac{\sigma^2}{2}p(Y_t|Y_{t-1})] = 0$$

where $p(Y_t|Y_{t-1}) = \lim_{t\to\infty} p(Y_t|Y_{t-1}, t)$. Then, there must be a constant $C$ s.t.

$$\frac{\sigma^2}{2} * \frac{\partial}{\partial Y}p(Y_t|Y_{t-1}) - \kappa(\mu - Y_t)p(Y_t|Y_{t-1}) = C.$$

We take an integral for both sides of the above equation on the real line and obtain:

$$\left|\int_{-\infty}^{\infty} CdY_t\right| = \left|\int_{-\infty}^{\infty}(\frac{\sigma^2}{2}\frac{\partial}{\partial Y}p(Y_t|Y_{t-1}) - \kappa(\mu - Y_t)p(Y_t|Y_{t-1}))dY_t\right|$$

$$\leq \kappa\mu\int_{-\infty}^{\infty} p(Y_t|Y_{t-1})dY_t + \kappa\left|\int_{-\infty}^{\infty} Y_t p(Y_t|Y_{t-1})dY_t\right| + \frac{\sigma^2}{2}\left|[p(Y_t|Y_{t-1})]_{-\infty}^{\infty}\right|$$

$$< \infty$$

Therefore, $C$ must be zero. This gives us a differential equation for the stationary transitional density:

$$\frac{\partial}{\partial Y}p(Y_t|Y_{t-1}) = \frac{2\kappa}{\sigma^2}(\mu - Y_t)p(Y_t|Y_{t-1}).$$

We have $\frac{\partial}{\partial Y}p(Y_t|Y_{t-1}) > 0$ if $Y_t < \mu$, and $\frac{\partial}{\partial Y}p(Y_t|Y_{t-1}) < 0$ if $Y_t > \mu$. This means $p(Y_t|Y_{t-1})$ has a unique maximum. $\qquad\square$

SHOWCASES

**Density on different time points for OU-1:**

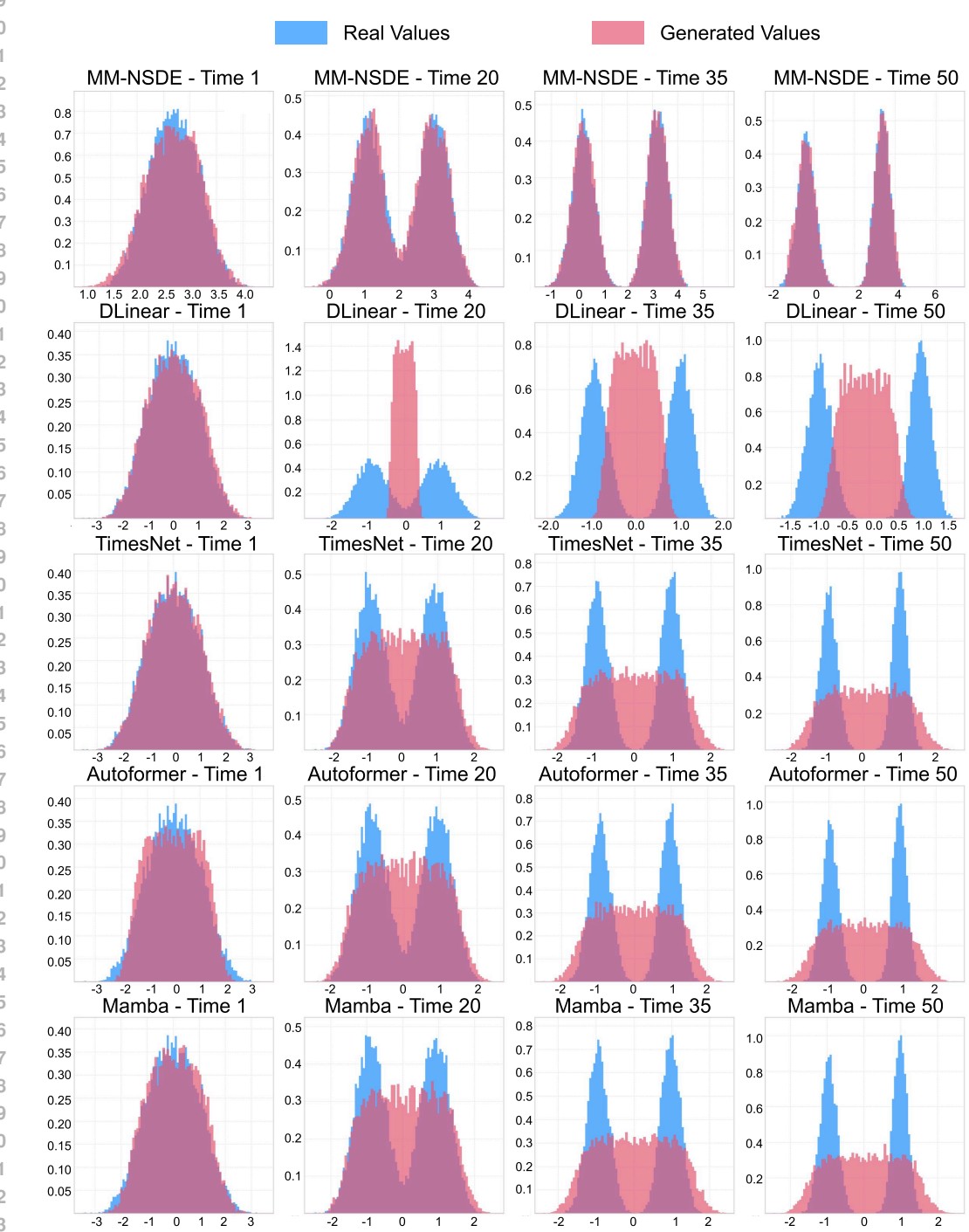

Figure 9: Fitting results of different models for OU-1. Red bins represent the density output by the models, while blue bins represent the true density of the data. Each row corresponds to a different model, and each column represents a specific time point.

**Density on different time points for GBM→GBM or CIR:**

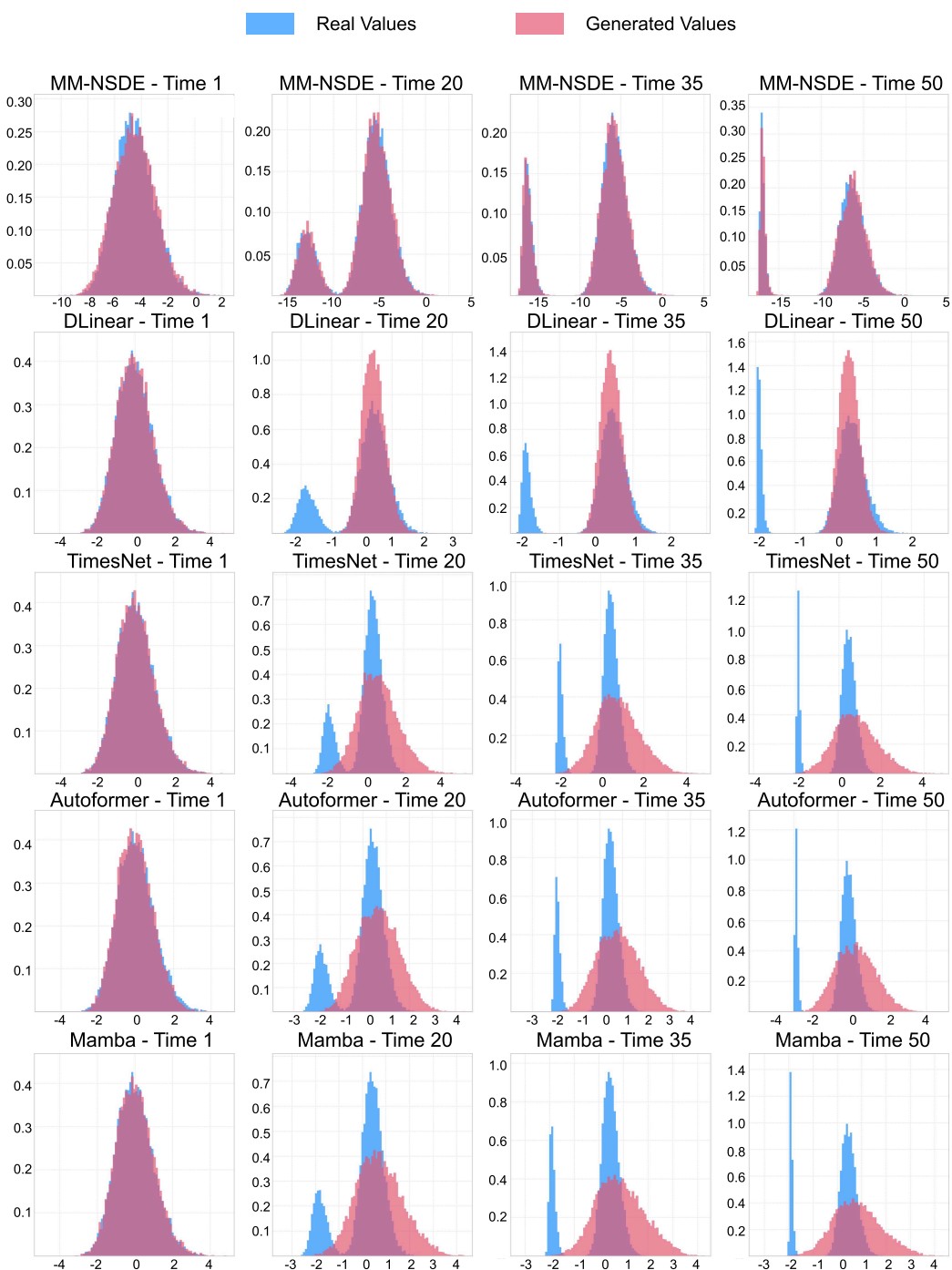

Figure 10: Fitting results of different models for GBM→GBM or CIR. Red bins represent the density output by the models, while blue bins represent the true density of the data. Each row corresponds to a different model, and each column represents a specific time point.

