# OpenReview forum: "State Aware Neural Stochastic Differential Equations for Multi-Modal Dynamics"
_ICLR.cc/2026/Conference — ICLR 2026 Conference Withdrawn Submission_

### Official Review · Reviewer_NGiw · 2025-10-25

**Soundness:** 2
**Presentation:** 2
**Contribution:** 2
**Rating:** 2
**Confidence:** 3

**Summary:**

The authors address the challenge of multimodality in Neural Stochastic Differential Equations (NSDEs). They argue that NSDEs are inherently limited in capturing multimodal distributions, supporting this claim with a short theoretical discussion (Theorem 3). To overcome this limitation, they introduce a latent variable $h_t$ that evolves as a function of multiple past observations $y_t$ and the past latent state. This latent variable modulates the original drift and diffusion terms multiplicatively. The model is trained using a Sinkhorn-based loss and evaluated on several real-world datasets.

**Strengths:**

Provides an interesting theoretical motivation for extending NSDEs.

Tackles the important problem of modeling multimodal systems.

**Weaknesses:**

The problem setup is not fully convincing. In my experience, NSDEs can generate multimodal distributions when trained via maximum likelihood using the Euler–Maruyama discretization (the “discretize-then-solve” approach). The authors cite Rakinsky, who discussed this approach. Using this method the one-step transition is a Gaussian and the model can be trained vie Maximum-Likelihood.

The proposed latent variable introduces non-Markovian dynamics, which seems to contradict the original NSDE formulation and motivation.

The choice of Sinkhorn loss is unusual in this context, and it is unclear how much of the performance stems from the new model versus the loss function itself.

The experimental evaluation is limited: no high-dimensional or challenging datasets (e.g., images) are considered. This weakens the empirical evidence, though this is a minor weakness compared to the theoretical concerns above.

**Questions:**

Major (critical for acceptance --> I will raise my score  if resolved):

Can you design a simple one-dimensional multimodal toy problem with known dynamics (ideally a well-specified SDE), and compare your method against a baseline NSDE trained via maximum likelihood with Euler–Maruyama discretization? This would clarify whether the multimodality issue truly exists.

Can you ablate the contributions of the proposed model and the Sinkhorn loss separately, ideally on the above toy problem? Without such disentanglement, it is difficult to assess the source of improvements.

Minor:

Since your system is no longer Markovian, it would be valuable to test a variant with a Markovian latent to better understand the trade-offs introduced by the non-Markovian design.

---

### Official Review · Reviewer_tky2 · 2025-10-27

**Soundness:** 2
**Presentation:** 3
**Contribution:** 2
**Rating:** 4
**Confidence:** 4

**Summary:**

The authors combine state space models and SDE-based forecasters to model dynamical systems. The authors improve the widely applied NSDEs to include a state space that captures the multi-modal nature of some particular dynamics.

**Strengths:**

1. The paper is well written with a clean presentation. The details are well explained.
2. The paper shows strong empirical performance on their experiments.

**Weaknesses:**

Minor:
1. The related work lags behind; there are quite a lot of NSDEs in recent years, for example [1], [2] to list a few, especially with [2] showing a multi-modal perspective through McKean-Vlasov SDE. Similarly, there are many newly proposed methods for sequential models as well. The author should have done a more thorough literature review.
2. The author should also address the setup similarities to a jump process, which is also extensively studied.

Major:
1. There seems to be a lack of motivation to apply a state-space model when the states are continuous to model multi-modal distribution in observation.
2. Some ablation studies is necessary to justify the improved performance of MM-NSDEs.

[1] Oh, YongKyung, Dong-Young Lim, and Sungil Kim. "Stable neural stochastic differential equations in analyzing irregular time series data." arXiv preprint arXiv:2402.14989 (2024).
[2] Yang, Haoming, Ali Hasan, Yuting Ng, and Vahid Tarokh. "Neural mckean-vlasov processes: Distributional dependence in diffusion processes." In International Conference on Artificial Intelligence and Statistics, pp. 262-270. PMLR, 2024.

**Questions:**

1. What is the input to $\tilde{f}_t(y)$ and $\tilde{g}_t(y)$? Is $y$ the only input, or is $t$ also an input?
2. Is the training objective of Sinkhorn divergence only applied to MM-NSDE? Is the performance gain obtained from this new loss function? What is the complexity of evaluating this loss function compared to other losses like MMD?
3. The real-world data doesn't seem like the standard evaluation dataset. Can the author run experiments similar to latent-SDE/Gan-SDE or more traditional TS datasets like exchange/Etth? The reviewer understands that the focus is on multi-modality and stochastic switching processes, but those are more synthetic rather than realistic, and don't provide a fair comparison to the other methods.
4. Why are the parameters for the synthetic dataset all relatively small (Table 4)?
5. Most of the experiments are single-dimensional. How does the model behave when it encounters more than 2 dimensions?
6. How are the continuous states related to the different modalities? Do the states themselves split into two modalities? It seems like the states follow a continuous ODE, so how does the multiplication of two continuous functions lead to the capability of modeling switching behavior in the data?
7. Related to the previous question, aside from the model performing well for time marginal distributions, do the sampled trajectories actually show that they switch between modes? Can the author show some predicted paths vs ground truth paths?

---

### Official Review · Reviewer_cXeg · 2025-11-01

**Soundness:** 2
**Presentation:** 3
**Contribution:** 2
**Rating:** 4
**Confidence:** 4

**Summary:**

The authors introduce a neural SDE model where the drift and diffusion functions are parameterized such that they promote multimodal behavior. The authors argue that standard neural networks parameterizing the drift and diffusion functions do not permit effective learning of multimodal distributions. The authors propose a regime switching SDE which selects the appropriately Lipschitz constrained drift and diffusion for the particular sampling task. This is to allow a balance between expressivity and smoothness. The authors finally devise an architecture that achieves this and then illustrate the performance on a number of empirical tasks. The results suggest an improvement in performance compared to baselines.

**Strengths:**

The authors have natural motivation for the proposed architecture.

The results are empirically sound and suggest a significant improvement upon existing methods.

The authors make an interesting case for training with MMD objective rather than others.

**Weaknesses:**

The constructions are arbitrary for how the regime switching is handled. One could argue that a sufficiently expressive network could also learn something similar.

The theoretical motivation is not very well connected within the presentation to the main architecture. It would be nice if this could be more clearly connected and motivated.

Figure 2 is not very clear or convincing when comparing NSDE to the MM-NSDE. Assuming that $x_t$ is actually supposed to be $Y_t$ we can think of a mapping within $f, g$ that has the same behavior as the MM-NSDE since $h_t$ depends on $Y_t$. This should possibly be clarified otherwise the construction is not well-motivated.

**Questions:**

Can the authors highlight in a bit more detail what it is specifically about the architecture of the process that is unique? Specifically, why the element wise multiplication by A and the correction term of B? It seems like there are a variety of ways to write something like this, and it would be helpful if the authors can connect it closer to the theoretical motivation.

Could, for example, an attention based network choose the correct regime based on the context? More broadly can the authors explain why existing architectures cannot represent these processes?

Is the diffusion function somehow constrained? It seems like it would be difficult to get stable training of the diffusion function in the current setup since this can drastically affect the behavior of the SDE, without adding additional constraints.

---

### Official Review · Reviewer_Ge9L · 2025-11-01

**Soundness:** 2
**Presentation:** 2
**Contribution:** 2
**Rating:** 6
**Confidence:** 5

**Summary:**

The paper “State-Aware Neural Stochastic Differential Equations for Multi-Modal Dynamics” introduces MM-NSDE, a variant of Neural SDEs designed to model multimodal stochastic processes. The authors diagnose a key limitation of standard NSDEs: a Lipschitz conflict between stability and expressiveness that prevents capturing multimodal transitions arising from shifts in the data-generating process (DGP). MM-NSDE addresses this by introducing a state-awareness module that infers latent regimes and a state-adaptive module that modulates drift and diffusion accordingly. The model employs an entropically regularized Sinkhorn divergence as the training objective for sensitivity to multimodality. Extensive experiments on simulated and real-world datasets (financial, environmental, and crypto) demonstrate that MM-NSDE significantly outperforms deterministic sequence models and prior SDE frameworks, while being computationally efficient.

**Strengths:**

The paper rigorously identifies and formalizes the Lipschitz–expressivity trade-off, supported by both theoretical analysis (Theorem 3) and empirical validation of multimodality thresholds. Additionally, the proposed state-aware adaptation mechanism is conceptually elegant and aligns neatly with the stochastic interpretation of DGP-switching. Empirically, MM-NSDE consistently surpasses baselines, including Mamba and Latent-SDE, across diverse synthetic and real datasets, with meticulous ablation and sensitivity studies further strengthening the empirical claims. Moreover, comprehensive theoretical derivations and thorough reproducibility documentation provided in the appendix demonstrate the authors' methodological rigor.

**Weaknesses:**

Despite its intuitive appeal, the theoretical underpinning of the "state-awareness" mechanism is somewhat limited, lacking a robust formal grounding as a stochastic process on latent manifolds; the latent dynamics appear heuristic rather than rigorously derived from stochastic differential equations. Moreover, conceptual novelty is ambiguous, given the substantial overlap with existing latent-regime and switching-state models; greater clarity distinguishing this work from prior approaches would strengthen the paper. Additionally, the benchmarks neglect important modern continuous-time diffusion or flow models, leaving out relevant recent score-based and neural-flow approaches. Finally, the paper exhibits excessive empirical focus while experiments are thorough, the theoretical contributions, particularly beyond Theorems 3 and 5, appear incremental rather than foundational.

**Questions:**

1. Could you clarify the empirical tightness of the inequality from Theorem 3, specifically regarding whether the observed onset of multimodality closely aligns with the theoretical predictions or if there's a noticeable empirical gap?

2. Given the centrality of the Lipschitz conflict, have you explored explicit Lipschitz regularization techniques, such as spectral penalties or adaptive scaling, as controllable trade-off parameters during training? If so, how does this explicit regularization compare to the implicit trade-off balancing achieved by the state-adaptive modules?

3. Could you provide a systematic analysis or visualization (e.g., a stability versus multimodality score curve) illustrating how transitions between stability and expressivity occur as the Lipschitz constants vary?

4. Do you anticipate a similar Lipschitz–expressivity trade-off bound extending naturally to other continuous-time stochastic models, such as neural diffusions or score-based SDEs, and could the adaptive mechanism proposed here effectively mitigate this dilemma in those contexts as well?

---

### Note · Authors · 2025-11-29

I have read and agree with the venue's withdrawal policy on behalf of myself and my co-authors.